# GUIDE: GATED UNCERTAINTY-INFORMED DISENTANGLED EXPERTS FOR LONG-TAILED RECOGNITION

**Yuan Dong**[1]*, **Zhe Zhao**[1,3]†, **Liheng Yu**[1], **Di Wu**[1], **Pengkun Wang**[1,2]†

[1]University of Science and Technology of China (USTC), Hefei 230026, China
[2]Suzhou Institute for Advanced Research, USTC, Suzhou 215123, China
[3]City University of Hong Kong
dongyuan01221@gmail.com, {zz4543, yuliheng, wdcxy}@mail.ustc.edu.cn
pengkun@ustc.edu.cn

## ABSTRACT

Long-Tailed Recognition (LTR) remains a significant challenge in deep learning. While multi-expert architectures are a prominent paradigm, we argue that their efficacy is fundamentally limited by a series of deeply entangled problems at the levels of representation, policy, and optimization. These entanglements induce homogeneity collapse among experts, suboptimal dynamic adjustments, and unstable meta-learning. In this paper, we introduce GUIDE, a novel framework conceived from the philosophy of Hierarchical Disentanglement. We systematically address these issues at three distinct levels. First, we disentangle expert representations and decisions through competitive specialization objectives to foster genuine diversity. Second, we disentangle policy-making from ambiguous signals by using online uncertainty decomposition to guide a dynamic expert refinement module, enabling a differentiated response to model ignorance versus data ambiguity. Third, we disentangle the optimization of the main task and the meta-policy via a two-timescale update mechanism, ensuring stable convergence. Extensive experiments on five challenging LTR benchmarks, including ImageNet-LT, iNaturalist 2018, CIFAR-100-LT, CIFAR-10-LT and Places-LT, demonstrate that GUIDE establishes a new state of the art, validating the efficacy of our disentanglement approach.

## 1 INTRODUCTION

Multi-expert architectures are the state-of-the-art paradigm in Long-Tailed Recognition (LTR), consistently advancing performance on real-world datasets with power-law class distributions (Wang et al., 2021; Zhang et al., 2022; Sanchez Aimar et al., 2023; Liu et al., 2019). The core intuition is that a committee of specialized experts can collectively provide more robust coverage of the entire data distribution, from head to tail (Shazeer et al., 2017; Jacobs et al., 1991).

However, despite empirical success, this paradigm is approaching a fundamental ceiling. We argue that progress is saturating due to a deep, unaddressed issue: a dependency chain of entangled learning processes that creates systemic performance bottlenecks. This chain begins with a foundational failure in representation learning that propagates to cripple the system's adaptive policy and destabilize its optimization.

The foundational problem is *representation-decision entanglement*. Existing methods often induce diversity through indirect means at the decision level, for instance using varied logit adjustments (Menon et al., 2021; Cao et al., 2019). Crucially, they do not explicitly decouple the representation learning process itself from the dominating influence of head classes (Kang et al., 2020). This allows powerful gradients from these classes to foster a homogeneity collapse, where all experts converge towards a similar, head-centric feature space (Kornblith et al., 2019; Morcos et al., 2018; Shen et al., 2015). *This collapse renders specialization ineffective and forms the root of subsequent failures.*

---

*Work done during a research internship at USTC-DILab.
†Corresponding authors.

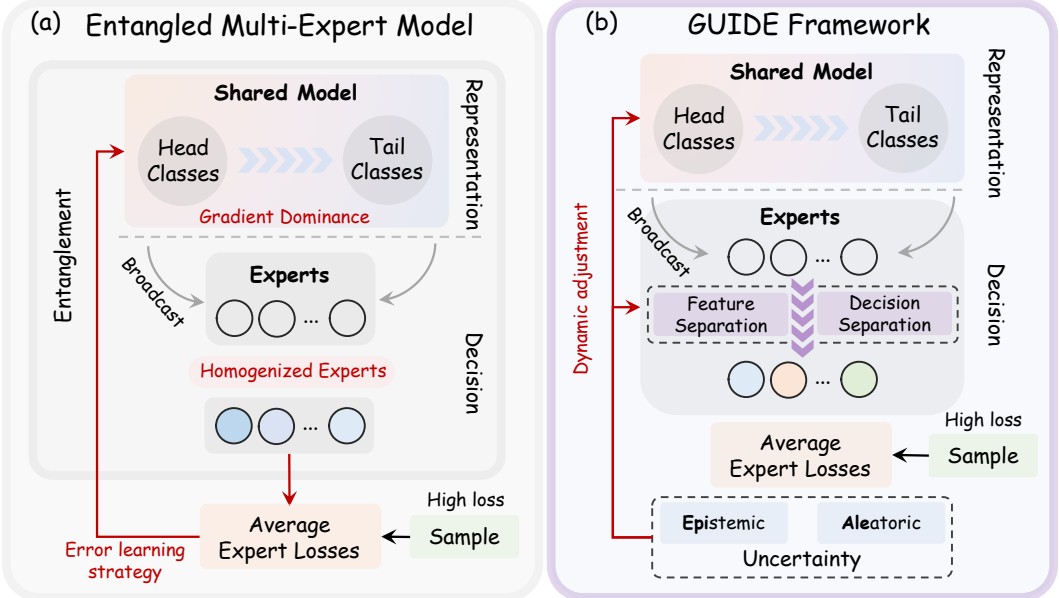

Figure 1: **(a) Conventional entangled multi-expert systems** suffer from homogeneity collapse due to shared representations and simplistic error-driven adaptation, leading to a flawed, cause-agnostic handling of difficult samples. **(b) Our GUIDE framework** breaks this entanglement by enforcing explicit feature and decision separation. This enables a principled, diagnostic approach where an epistemic / aleatoric uncertainty analysis guides the targeted allocation of learning resources.

This representational collapse directly compromises the system's adaptive capabilities, leading to *cause-symptom entanglement*. A committee of functionally similar experts lacks the diverse perspectives needed for a reliable diagnosis of difficult samples. Advanced LTR methods thus rely on ambiguous signals like high training loss, conflating the symptom with its cause. This cause can be model ignorance (*epistemic uncertainty*) or inherent data ambiguity (*aleatoric uncertainty*) (Kendall & Gal, 2017; Lakshminarayanan et al., 2017; Shen et al., 2018). Without genuine expert diversity, distinguishing these becomes unreliable, leading to a chronic misallocation of model resources.

Finally, the instability at the representation and policy levels creates a volatile optimization landscape, culminating in *learning-meta-learning entanglement*. The slow, deliberate optimization of an adaptive strategy is fundamentally at odds with the fast, high-variance optimization of the primary recognition task. Since this primary task already operates on collapsed representations and is guided by a flawed adaptive policy, its high-variance gradients overwhelm the subtle updates required to refine the meta-policy. *This pervasive instability prevents the system from converging to a robust and effective self-organizing strategy* (Heusel et al., 2017).

To resolve these interconnected entanglements in a unified manner, we propose **GUIDE**, a Gated Uncertainty-Informed Disentangled Experts framework built upon the principle of *hierarchical disentanglement*. As illustrated in Figure 1, GUIDE re-architects the learning process to systematically decouple these coupled mechanisms, addressing each level of the dependency chain in order.

▶ First, to establish a stable foundation at the `representation` level, we introduce competitive specialization objectives. This active pressure forces experts into distinct functional niches by decorrelating features and maximizing predictive divergence (Krogh & Vedelsby, 1994; Liu & Yao, 1999). The approach is theoretically grounded in tightening the ensemble's loss bound and effectively prevents homogeneity collapse (Wang et al., 2017; 2025). ▶ Building on this foundation of diverse experts, GUIDE can perform reliable diagnosis at the `policy` level. We shift from reactive adaptation to principled, diagnostic intervention. Our framework learns a differentiated policy by decomposing predictive uncertainty into its epistemic and aleatoric components. A dynamic expert refinement module then allocates refinement capacity based on a robust assessment of model igno-

rance, ensuring targeted learning (Gao et al., 2017). ▶ Finally, the stability from the disentangled representation and policy levels creates a protected optimization landscape. At the `optimization` level, a two-timescale update rule decouples the fast primary learning from the slow meta-learning. This ensures the stable convergence of strategic adjustments (Heusel et al., 2017).

We summarize our contributions as follows:

❶ *New problem and insight*: we identify a dependency chain of fundamental entanglement problems in long-tailed recognition and propose GUIDE, a novel hierarchical disentanglement framework to systematically resolve it.

❷ *New learning framework*: we design a suite of three interconnected mechanisms: a competitive specialization framework to build a diverse representational foundation, an uncertainty-guided adaptation policy enabled by this diversity, and a two-timescale optimization process made stable by the preceding levels.

❸ *Compelling empirical results*: extensive experiments on five challenging long-tailed benchmarks, ImageNet-LT, iNaturalist 2018, CIFAR-100-LT, CIFAR-10-LT and Places-LT, establish a new state of the art for GUIDE, validating our systemic disentanglement approach.

## 2 RELATED WORK

▶ **Multi-Expert Long-Tailed Learning and Representation Collapse.** Multi-expert LTR methods (Wang et al., 2021; Zhang et al., 2022; Sanchez Aimar et al., 2023) encourage diversity indirectly at the decision level, leaving representations vulnerable to *homogeneity collapse*. This *representation-decision entanglement*, driven by head-class gradients, forces experts into correlated feature spaces and limits their collective capacity. GUIDE directly resolves this by enforcing competitive specialization at both feature and decision levels, simultaneously decorrelating features and maximizing predictive divergence to learn complementary experts. Relatedly, MDCS (Zhao et al., 2023) also promotes expert diversity using consistency self-distillation, though our hierarchical approach addresses a broader cascade of entanglements, from representation to optimization.

▶ **Uncertainty-Guided Adaptation.** Adaptive LTR methods often suffer from *cause-symptom entanglement*, reacting to ambiguous signals like high training loss with suboptimal, one-size-fits-all adjustments. While prior work uses uncertainty for auxiliary tasks (Chen & Su, 2023; Yang et al., 2024), GUIDE is the first to our knowledge to leverage decomposed epistemic (model) and aleatoric (data) uncertainty for a *differentiated, structural* adaptation policy. This enables targeted refinement for model ignorance (high epistemic) while promoting robustness against data noise (high aleatoric).

▶ **Meta-Optimization in LTR.** Meta-learning for LTR (Ren et al., 2018; Shu et al., 2019) faces optimization instability due to interference between the fast primary task and the slow meta-policy optimization. This *learning and meta-learning entanglement* hinders convergence. GUIDE resolves this using a Two-Timescale Stochastic Approximation (TTSA) scheme (Borkar, 1997), using differential learning rates to decouple the two optimization loops and ensure stable policy convergence.

## 3 GUIDE: AN EFFECTIVE HIERARCHICAL DISENTANGLEMENT METHOD

The GUIDE framework is designed to systematically resolve the entanglement problems in long-tailed learning through a Hierarchical Disentanglement approach. We decompose the learning process into three distinct levels, each targeting a specific entanglement in a sequential manner that respects their underlying dependencies. The overall architecture is illustrated in Figure 2. Below, we detail the mechanisms at each level, motivating their design and explaining their synergistic roles. Our framework is built upon a principled design, theoretically motivated by established concepts in ensemble learning and optimization theory. Rather than proposing new foundational theorems, our contribution lies in demonstrating how a synergistic application of these principles can resolve the core entanglements in LTR, with our design choices being validated through extensive empirical analysis.

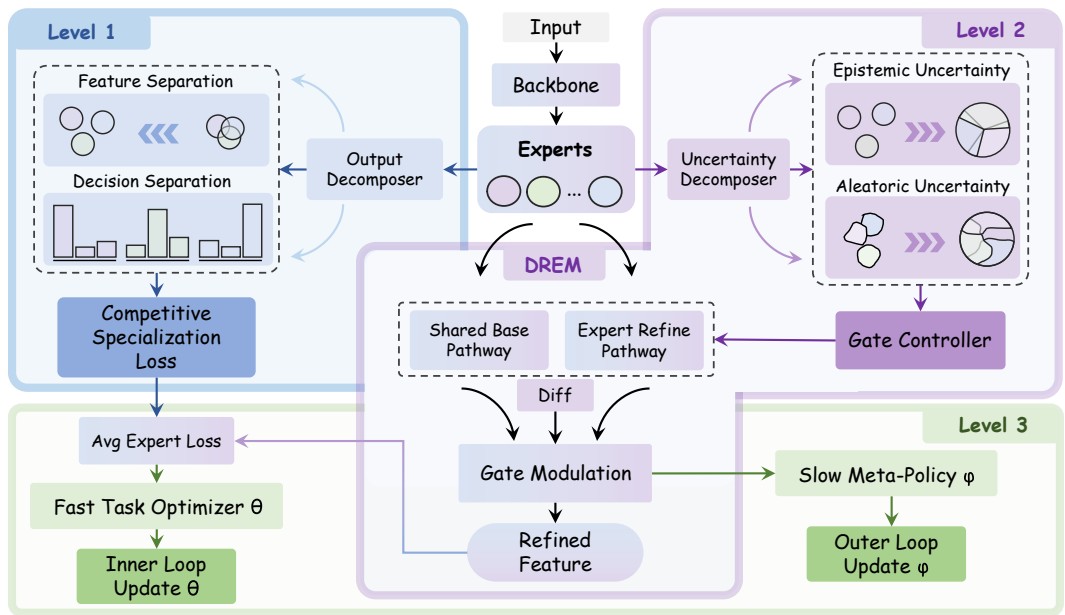

Figure 2: The Hierarchical Disentanglement architecture of GUIDE. An input is processed via a shared backbone and a committee of experts. **Level ❶** enforces diversity by penalizing feature and decision overlap. This enables **Level ❷**, where expert disagreement is decomposed into epistemic (model) and aleatoric (data) uncertainty. These signals drive a gate controller to modulate the Dynamic Expert Refinement Module (DERM). Finally, **Level ❸** decouples optimization into a fast inner loop for task parameters ($\theta$) and a slow outer loop for the meta-policy ($\phi$), which closes the meta-learning loop by updating the gate controller.

## 3.1 LEVEL ❶: REPRESENTATION DISENTANGLEMENT VIA COMPETITIVE SPECIALIZATION

The foundational level of our framework confronts the *representation-decision entanglement* causing homogeneity collapse. In standard multi-expert systems, head-class gradients dominate, forcing all experts into a functionally redundant, head-centric feature space. Our core insight is that genuine diversity cannot be a byproduct; it must be an explicit optimization objective. We introduce a paradigm of **competitive specialization**, establishing a dynamic equilibrium where experts collaborate while competing for distinct functional niches. This design is guided by Theorem 1.

**Theorem 1** (Diversity-driven Bound Tightening). *For an ensemble of E experts, (a) the ensemble's negative log-likelihood (NLL) is upper-bounded by the average of individual expert NLLs. (b) This performance gap is directly related to the predictive diversity among experts, which can be measured and optimized by maximizing the Jensen-Shannon Divergence (JSD).*

A formal statement and proof are provided in Appendix A.2. Theorem 1 connects the ensemble's performance bound to its predictive diversity. While this bound is not an equality, it provides a strong theoretical motivation to use JSD, a standard measure of distribution diversity, as an effective proxy objective for encouraging expert specialization. Motivated by this principle, our Level ❶ objective integrates a collaborative loss with two competitive regularizers. The main collaborative task is driven by a standard cross-entropy loss, $\mathcal{L}_{\text{main}}$, on aggregated, logit-adjusted predictions. For a given input $x$ with true label $y$, the main loss over an ensemble of $E$ experts is:

$$\mathcal{L}_{\text{main}} = -\log \frac{\exp\left(\sum_{e=1}^{E} \tilde{z}_e(x)_y\right)}{\sum_{c=1}^{C} \exp\left(\sum_{e=1}^{E} \tilde{z}_e(x)_c\right)}. \tag{1}$$

Here, the adjusted logit for class $c$ from expert $e$ is $\tilde{z}_e(x)_c = z_e(x)_c + \tau_e \log \pi_{\text{train}}(c)$, where $z_e(x)$ is the raw logit vector, $\pi_{\text{train}}(c)$ is the training prior, and $\tau_e$ is an expert-specific hyperparameter. Although this loss promotes collaboration, it is insufficient to prevent homogeneity collapse, so we

introduce two synergistic regularizers to actively enforce diversity. For **representation diversity**, we minimize the cosine similarity between the feature vectors extracted by different experts:

$$\mathcal{L}_{\text{decouple}} = \frac{2}{E(E-1)} \sum_{1 \leq i < j \leq E} \frac{f_i(x)^\top f_j(x)}{\|f_i(x)\|_2 \cdot \|f_j(x)\|_2 + \varepsilon}. \tag{2}$$

where $f_i(x)$ and $f_j(x)$ are feature vectors from experts $i$ and $j$, and $\varepsilon$ is a small constant for numerical stability. For **prediction diversity**, we explicitly maximize the JSD of the predictive distributions from the experts. The final Level ❶ objective orchestrates these collaborative and competitive forces:

$$\mathcal{L}_{\text{total}}^{(1)} = \mathcal{L}_{\text{main}} + \lambda_{\text{dec}}\mathcal{L}_{\text{decouple}} - \lambda_{\text{div}}\text{JSD}\big(\{p_{e,T}(\cdot|x)\}\big). \tag{3}$$

The terms $\lambda_{\text{dec}}$ and $\lambda_{\text{div}}$ are scalar weights that balance the three objectives, and $p_{e,T}(\cdot|x)$ is the temperature-scaled predictive distribution from expert $e$. By orchestrating these forces, Level ❶ resolves the representation-decision entanglement. Its immediate effect is the cultivation of a truly diverse ensemble. This diversity is the critical precondition for Level ❷, as it transforms the disagreement among experts from noise into a reliable, high-fidelity diagnostic signal.

## 3.2 LEVEL ❷: POLICY DISENTANGLEMENT VIA ADAPTIVE META-LEARNING

Capitalizing on the functionally diverse ensemble forged in Level ❶, we can now effectively address the *cause-symptom entanglement*. This problem arises from reacting to ambiguous signals like high loss, which conflates model ignorance (*epistemic uncertainty*) with data ambiguity (*aleatoric uncertainty*). Our solution is a principled, two-stage process: first **diagnose** the true cause of difficulty, then execute a **targeted** intervention, with behavior guaranteed by Theorem 2.

**Theorem 2** (Policy Monotonicity). *The refinement strength of the Dynamic Expert Refinement Module (DERM) for a given class is a monotonically increasing function with respect to model's epistemic uncertainty and a monotonically decreasing function with respect to its aleatoric uncertainty.*

A proof is provided in Appendix A.2. Our implementation first performs **online diagnosis**. To achieve this, we leverage the high-fidelity signal from our diverse experts and decompose the predictive uncertainty by adopting a widely-used formulation from Bayesian deep learning (Kendall & Gal, 2017; Lakshminarayanan et al., 2017). This approach provides a clear and empirically effective distinction between model ignorance (epistemic) and data ambiguity (aleatoric), which is critical for our targeted intervention and is validated in Section 4.3:

$$\text{Aleatoric Uncertainty (Ale):} \quad \text{Ale}_T(x) = \frac{1}{E} \sum_{e=1}^{E} H\big(p_{e,T}(\cdot|x)\big) \tag{4}$$

$$\text{Epistemic Uncertainty (Epi):} \quad \text{Epi}_T(x) = H\big(\bar{p}_T(\cdot|x)\big) - \text{Ale}_T(x). \tag{5}$$

Here, $H(\cdot)$ denotes the Shannon entropy and $\bar{p}_T(\cdot|x)$ is the average predictive distribution. Subsequently, the **DERM** executes the targeted policy. It consists of a shared **foundation pathway**, $F_{\text{found}}$, and expert-specific **refinement pathways**, $F_{\text{refine},e}$, which are neural network modules designed to process the features extracted by the backbone. For a given class $c$, these are combined via an **adaptive residual mixture**:

$$\mathbf{f}_e(x;c) = F_{\text{found}}(x) + g_{e,c} \cdot \left(F_{\text{refine},e}(F_{\text{found}}(x)) - F_{\text{found}}(x)\right). \tag{6}$$

The **refinement strength gate** $g_{e,c}$ is a learnable function of stable, class-level Exponential Moving Averages of the diagnosed uncertainties, denoted as $\bar{\text{Epi}}_{T,c}^{(t)}$ and $\bar{\text{Ale}}_{T,c}^{(t)}$ for class $c$ at step $t$:

$$\tilde{g}_{e,c} = \alpha_e \cdot \bar{\text{Epi}}_{T,c}^{(t)} - \beta_e \cdot \bar{\text{Ale}}_{T,c}^{(t)} + \gamma_e \tag{7}$$

$$g_{e,c} = \sigma(\tilde{g}_{e,c}) \cdot (g_{\max} - g_{\min}) + g_{\min}. \tag{8}$$

In this formulation, $\{\alpha_e, \beta_e, \gamma_e\}_e$ are learnable parameters, $\sigma(\cdot)$ is the sigmoid function, and $g_{\min}, g_{\max}$ are hyperparameters defining the gate's range. This mechanism resolves the cause-symptom entanglement. Its effect is to replace chaotic, error-driven reactions with a principled allocation of model capacity. This targeted learning process inherently stabilizes the training dynamics, creating a more tractable optimization landscape, which is the essential prerequisite for Level ❸ to successfully learn a robust meta-policy.

### 3.3 LEVEL ❸: OPTIMIZATION DISENTANGLEMENT VIA TWO-TIMESCALE UPDATES

Operating within the stabilized learning environment established by the preceding levels, we can finally confront the *learning-meta-learning entanglement*. This conflict occurs because the slow, deliberate optimization of the adaptive meta-policy is easily overwhelmed by the fast, high-variance optimization of the primary task. Our solution is to explicitly decouple these processes by assigning them different timescales, a strategy grounded in Two-Timescale Stochastic Approximation (TTSA), as formalized in Proposition 1.

**Proposition 1** (Satisfaction of TTSA Conditions). *By partitioning the model parameters into fast variables $\theta$ and slow variables $\phi$, and employing an update scheme where their respective learning rates satisfy $\eta_\phi \ll \eta_\theta$, our optimization process constitutes a valid TTSA algorithm.*

A discussion is provided in Appendix A.2. We realize this principle by partitioning the model's learnable parameters based on their function and required update speed. This results in two distinct sets of variables:

▶ *Fast Variables* $\theta$: Main network parameters for feature extraction and recognition, including the shared backbone and expert-specific DERM pathways ($F_{\text{found}}$, $F_{\text{refine},e}$).

▶ *Slow Variables* $\phi$: The parameters $\{\alpha_e, \beta_e, \gamma_e\}_e$ governing the *Strategy Controllers*, which constitute the adaptive meta-policy.

These sets are updated on their respective timescales. The ***inner loop (fast timescale)*** updates the main network parameters $\theta$ at each training step $k$ with a learning rate $\eta_\theta$:

$$\theta_{k+1} = \theta_k - \eta_\theta \nabla_\theta \mathcal{L}_{\text{GUIDE}}(x_k, y_k; \theta_k, \phi_t). \tag{9}$$

Conversely, the ***outer loop (slow timescale)*** updates the meta-policy controllers $\phi$ less frequently (every epoch $t$) with a smaller learning rate $\eta_\phi$, optimizing performance on a validation set $\mathcal{V}$:

$$\phi_{t+1} = \phi_t - \eta_\phi \nabla_\phi \mathbb{E}_{(x_v, y_v) \in \mathcal{V}}[\mathcal{L}_{\text{main}}(x_v, y_v; \theta_t, \phi_t)]. \tag{10}$$

This principled decoupling is the capstone of our hierarchical approach. Having first established a diverse ensemble and then a stable, diagnostic policy, the optimization landscape is now sufficiently well-behaved for a meta-learning algorithm to operate effectively. The effect of Level ❸ is to create a protected optimization channel, allowing the meta-policy to safely converge. By solving the entanglements in a specific, hierarchical order, GUIDE guides the entire framework towards a robust, self-organizing equilibrium, fully unlocking the potential of the multi-expert paradigm.

## 4 EXPERIMENTS

### 4.1 EXPERIMENTAL SETUPS

▶ **Datasets.** We validate our method on a suite of **five** widely-used benchmarks. Our primary analysis is conducted on four challenging datasets: CIFAR-100-LT (Cao et al., 2019), ImageNet-LT (Liu et al., 2019), Places-LT (Liu et al., 2019), and iNaturalist 2018 (Van Horn et al., 2018), which cover a wide range of scales and imbalance ratios. To further demonstrate general applicability, a full set of results on a fifth benchmark, CIFAR-10-LT (Cao et al., 2019), is provided in Appendix B.3. Detailed statistics for all five datasets are in Appendix B.1.

▶ **Baselines.** We compare GUIDE against a comprehensive set of state-of-the-art methods. Our comparison strategy is principled and transparent. We primarily report results from original publications to ensure fidelity. For methods lacking results on a relevant benchmark, we reproduced

Table 1: Comparison of Top-1 accuracy (%) with state-of-the-art methods. Dashed underline indicates results reproduced by us using the official codebases of the respective authors. Best results are in **bold**, second best are underlined.

| Method | CIFAR-100-LT | | | ImageNet-LT | iNaturalist 2018 | Places-LT |
|---|---|---|---|---|---|---|
| | IR=10 | IR=50 | IR=100 | | | |
| Softmax | 59.1 | 45.6 | 41.4 | 48.0 | 64.7 | 31.4 |
| LDAM-DRW (Cao et al., 2019) | 58.7 | 46.6 | 42.0 | 48.8 | 64.6 | 36.8 |
| Balanced Softmax (Ren et al., 2020) | 61.0 | 50.9 | 46.1 | 52.3 | 70.6 | 39.4 |
| LADE (Hong et al., 2021) | 61.6 | 50.1 | 45.6 | 52.3 | 69.3 | 39.2 |
| MiSLAS (Zhong et al., 2021) | 62.5 | 51.5 | 46.8 | 51.4 | 70.7 | 38.3 |
| RIDE (3 experts) (Wang et al., 2021) | 61.8 | 51.7 | 48.0 | 56.3 | 71.8 | 40.3 |
| SADE (Zhang et al., 2022) | 63.6 | 53.8 | 48.8 | 58.8 | 72.7 | 40.9 |
| BalPoE (Sanchez Aimar et al., 2023) | 64.8 | 56.3 | 52.0 | 59.3 | 75.0 | 40.8 |
| MDCS (Zhao et al., 2023) | - | 57.2 | 53.2 | 59.3 | 72.5 | **42.4** |
| LSC (Wei et al., 2024) | 65.0 | 56.5 | 51.8 | 60.2 | 73.9 | 41.3 |
| BCL (Zhu et al., 2022) | 64.9 | 56.6 | 51.9 | 56.0 | 71.8 | - |
| PRL (Zhao et al., 2024) | 65.6 | 57.3 | 52.8 | 60.8 | 75.1 | 41.6 |
| ProCo (Du et al., 2024) | 65.5 | 57.1 | 52.8 | 57.3 | 73.5 | - |
| ConCutmix (Pan et al., 2024) | 64.5 | 57.4 | 53.2 | 58.5 | 72.1 | - |
| LOS (Sun et al., 2025) | **69.7** | 58.8 | 54.9 | 54.4 | 70.8 | - |
| FeatRecon (Yi et al., 2025) | 65.3 | 57.0 | 52.5 | 56.8 | 72.9 | 41.3 |
| GUIDE | 69.2 | 60.3 | 56.4 | **62.5** | **76.1** | 42.2 |
| *Longer training* | | | | | | |
| PaCo (Cui et al., 2021) | 64.2 | 56.0 | 52.0 | 57.0 | 73.2 | - |
| SADE (Zhang et al., 2022) | 65.3 | 57.3 | 52.2 | - | 74.0 | 39.5 |
| NCL (Li et al., 2022) | - | 58.2 | 54.2 | 59.5 | 76.9 | - |
| BCL (Zhu et al., 2022) | 66.9 | 58.6 | 53.9 | 59.0 | 74.8 | - |
| BalPoE (Sanchez Aimar et al., 2023) | 68.1 | 60.1 | 55.9 | 59.7 | 76.9 | 41.1 |
| MDCS (Zhao et al., 2023) | - | 60.1 | 56.1 | 60.7 | 75.6 | - |
| ProCo (Du et al., 2024) | 67.8 | 58.9 | 54.2 | 58.5 | 75.6 | - |
| ConCutmix (Pan et al., 2024) | 68.3 | 59.5 | 55.1 | 58.9 | 75.7 | - |
| FeatRecon (Yi et al., 2025) | 67.2 | 58.5 | 54.0 | 59.3 | 75.9 | - |
| GUIDE | **70.2** | **62.3** | **57.7** | **63.4** | **77.8** | **43.1** |

Table 2: Detailed performance breakdown (%) on CIFAR-100-LT with ResNet-32. Our method shows substantial gains on Few-shot classes, addressing the core challenge of LTR.

| Method | CIFAR-100-LT (IR=100) | | | | CIFAR-100-LT (IR=50) | | | |
|---|---|---|---|---|---|---|---|---|
| | Many | Medium | Few | Overall | Many | Medium | Few | Overall |
| LDAM-DRW (Cao et al., 2019) | 61.2 | 39.5 | 14.4 | 42.0 | 64.9 | 45.1 | 28.7 | 46.6 |
| RIDE (3 experts) (Wang et al., 2021) | 64.7 | 46.5 | 22.3 | 48.0 | 67.5 | 50.8 | 27.9 | 51.7 |
| SADE(Zhang et al., 2022) | 63.4 | 49.2 | 25.1 | 49.8 | 67.2 | 53.5 | 31.8 | 53.9 |
| BalPoE (Sanchez Aimar et al., 2023) | 65.3 | 51.1 | 28.0 | 52.0 | 69.2 | 55.9 | 34.5 | 56.3 |
| PRL (Zhao et al., 2024) | 68.7 | 55.3 | 31.2 | 52.8 | 67.5 | 53.5 | 39.0 | 56.6 |
| GUIDE | **71.3** | **59.1** | **36.0** | **56.4** | 70.2 | 56.1 | 47.4 | 60.3 |
| GUIDE (longer) | **74.4** | **60.0** | **36.1** | **57.7** | 72.4 | 57.5 | 50.1 | 62.3 |

their performance using official codebases. These reproduced results are clearly marked in our main comparison table. A detailed justification for each reproduced result are provided in Appendix B.2.

▶ **Implementation and Evaluation Details.** To ensure a fair comparison, we employ standard backbones for all experiments: ResNet-32 for CIFAR-100-LT and CIFAR-10-LT, ResNet-50 for ImageNet-LT and iNaturalist 2018, and ResNet-152 for Places-LT. We report Top-1 accuracy on the official balanced test sets, supplemented by a fine-grained analysis on Many, Medium, and Few-shot splits. All models are trained from scratch using an SGD optimizer under both standard and longer training schedules. Full implementation details are provided in Appendix B.1.

Table 3: Top-1 accuracy (%) on CIFAR-100-LT (IR=100) with various unknown test class distributions. 'Prior' indicates whether the method uses the test class prior for post-hoc adjustment. All models are from the standard training schedule.

| Method | Prior | Forward-LT | | | | | Uni. | Backward-LT | | | | |
|---|---|---|---|---|---|---|---|---|---|---|---|---|
| | | 50 | 25 | 10 | 5 | 2 | 1 | 2 | 5 | 10 | 25 | 50 |
| Softmax | ✗ | 63.3 | 62.0 | 56.2 | 52.5 | 46.4 | 41.4 | 36.5 | 30.5 | 25.8 | 21.7 | 17.5 |
| MiSLAS (Zhong et al., 2021) | ✗ | 58.8 | 57.2 | 55.2 | 53.0 | 49.6 | 46.8 | 43.6 | 40.1 | 37.7 | 33.9 | 32.1 |
| LADE (Hong et al., 2021) | ✗ | 56.0 | 55.5 | 52.8 | 51.0 | 48.0 | 45.6 | 43.2 | 40.0 | 38.3 | 35.5 | 34.0 |
| LADE (Hong et al., 2021) | ✓ | 62.6 | 60.2 | 55.6 | 52.7 | 48.2 | 45.6 | 43.8 | 41.1 | 41.5 | 40.7 | 41.6 |
| RIDE (Wang et al., 2021) | ✗ | 63.0 | 59.9 | 57.0 | 53.6 | 49.4 | 48.0 | 42.5 | 38.1 | 35.4 | 31.6 | 29.2 |
| SADE (Zhang et al., 2022) | ✗ | 65.2 | 62.5 | 58.8 | 55.4 | 51.2 | 48.8 | 43.0 | 43.9 | 42.4 | 42.2 | 42.0 |
| BalPoE (Sanchez Aimar et al., 2023) | ✗ | 69.0 | 65.2 | 61.2 | 59.0 | 54.2 | 51.7 | 45.7 | 46.6 | 45.2 | 45.2 | 45.8 |
| LSC (Wei et al., 2024) | ✗ | 67.8 | 64.2 | 60.2 | 58.1 | 53.2 | 51.6 | 44.7 | 45.7 | 44.2 | 44.7 | 48.0 |
| PRL (Zhao et al., 2024) | ✗ | 69.5 | 65.7 | 61.7 | 59.5 | 54.7 | 52.2 | 46.2 | 47.1 | 45.7 | **45.7** | **48.5** |
| GUIDE | ✗ | 68.5 | **66.7** | **64.4** | **62.6** | **59.2** | **56.4** | **53.3** | **49.7** | **46.7** | 43.2 | 40.7 |
| GUIDE (longer) | ✗ | **71.9** | **70.0** | **67.3** | **64.7** | **60.8** | **57.7** | **54.4** | **50.6** | **47.7** | 43.6 | 41.5 |

## 4.2 COMPARISON WITH STATE-OF-THE-ART METHODS

▶ **Performance on Standard Benchmarks.** As shown in Table 1, GUIDE establishes a new state of the art on nearly all standard long-tailed benchmarks, including ImageNet-LT, iNaturalist 2018, and Places-LT, under both standard and longer training schedules. The performance gains are particularly pronounced on large-scale datasets, highlighting the scalability of our hierarchical disentanglement approach. The detailed breakdown on CIFAR-100-LT (Table 2) reveals the source of this advantage: GUIDE delivers significant performance gains in the challenging medium- and few-shot regimes, making notable progress on the core problem of LTR.

▶ **Robustness to Distribution Shift.** A critical goal of LTR is to learn representations that are not merely adapted to the training distribution but are fundamentally robust to real-world distribution shifts. We go beyond standard evaluation and assess this capability on a suite of unseen, skewed test distributions, following the protocol of Zhang et al. (2022) and Zhao et al. (2024). This setup, particularly the challenging Backward-LT distributions which completely invert the training class frequencies, serves as a strong proxy for out-of-distribution (OOD) robustness.

As detailed in Table 3, GUIDE demonstrates superior generalization under these shifts. While competitive on distributions that still favor head classes (Forward-LT), GUIDE's true advantage emerges in the most difficult scenarios. It surpasses all prior methods on the standard uniform test set and, more strikingly, establishes a significant performance margin on the OOD-like Backward-LT distributions. This strong performance under severe distribution shifts validates that our hierarchical disentanglement fosters a more fundamental and less biased understanding of tail classes, leading to enhanced robustness beyond the training prior.

## 4.3 ABLATION STUDIES, ANALYSIS, AND DISCUSSION

To rigorously and empirically validate our central thesis of *Hierarchical Disentanglement*, we conduct a series of in-depth ablation studies. Our ablation baseline, denoted as the Entangled Baseline, is a strong multi-expert model with three experts sharing a backbone, trained via a standard logit-adjusted cross-entropy loss and including no explicit disentanglement mechanisms. Collectively, our studies confirm the critical necessity of each proposed component and the framework's overall robustness. Beyond validating our framework's components, we also analyze the practical trade-offs of its inference strategies, demonstrating its flexibility for diverse real-world applications. See Appendix B.4 for more detailed analysis.

Table 4: Ablation study on the contribution of each disentanglement level. Performance consistently improves as each level of disentanglement is introduced, culminating in our final reported performance.

| ❶ | ❷ | ❸ | Many | Medium | Few | Overall |
|---|---|---|---|---|---|---|
| | Baseline | | 62.1 | 45.0 | 21.3 | 45.8 |
| ✓ | | | 64.9 | 51.1 | 26.0 | 50.4 |
| | ✓ | | 65.7 | 51.5 | 27.1 | 51.3 |
| | | ✓ | 64.2 | 49.8 | 25.5 | 49.9 |
| ✓ | ✓ | | 68.5 | 55.4 | 30.1 | 52.8 |
| ✓ | | ✓ | 67.3 | 53.8 | 28.9 | 52.1 |
| | ✓ | ✓ | 67.9 | 54.2 | 29.3 | 52.5 |
| | GUIDE | | **71.3** | **59.1** | **36.0** | **56.4** |

Table 5: Analysis of individual mechanisms. (Left) The synergistic effect of diversity losses in Level ❶. (Right) Comparison of different gating policies for the DERM in Level ❷.

| Level ❶: Diversity Losses | | Level ❷: Gating Policy | |
|---|---|---|---|
| Configuration | Accuracy (%) | Configuration | Accuracy (%) |
| Entangled Baseline | 45.8 | No Adaptation (gate=0.5) | 52.1 |
| + $\mathcal{L}_{\text{decouple}}$ only | 47.7 | Static (Inverse Freq.) | 53.6 |
| + $\mathcal{L}_{\text{div}}$ only | 47.5 | Agnostic (Total Uncert.) | 54.9 |
| + Both (Level ❶ Full) | **50.4** | **GUIDE Policy (Ours)** | **56.4** |

Table 6: Trade-off analysis for inference strategies. The two-step method consistently improves few-shot accuracy at a predictable latency cost across datasets of different scales. Speed is measured on an NVIDIA RTX 4090 GPU.

| Dataset | Inference Strategy | Efficiency | | Top-1 Accuracy (%) | |
|---|---|---|---|---|---|
| | | Latency (ms/img) | Throughput (img/s) | Overall | Few-shot |
| CIFAR-100-LT | Two-Step (Default) | 9.2 | 1054 | 56.4 | 36.0 |
| | Single-Pass | 5.1 | 1923 | 55.1 | 33.9 |
| ImageNet-LT | Two-Step (Default) | 25.6 | 391 | 62.5 | 47.6 |
| | Single-Pass | 14.1 | 709 | 60.7 | 45.2 |

▶ **The Necessity of Each Disentanglement Level.**

Our comprehensive ablation in Table 4 analyzes all component combinations, revealing their individual and synergistic contributions. First, each level demonstrates significant standalone efficacy, with Level ❶ and Level ❷ yielding the largest gains of $+4.6\%$ and $+5.5\%$ respectively. Second, all two-component combinations outperform any single component, showing their complementary nature. Notably, the combination of Level ❶ and ❷ is the strongest pair ($52.8\%$), underscoring a critical insight: a principled policy (Level ❷) operates most effectively on a diverse representational foundation (Level ❶). Finally, the full GUIDE model, integrating all three levels, attains a peak performance of $56.4\%$. This result significantly surpasses all partial configurations, demonstrating a powerful synergy that validates our central thesis: the framework's full potential is unlocked only by sequentially resolving all three entanglements.

▶ **Deconstructing the Disentanglement Mechanisms.** We further dissect the core mechanisms in Table 5. The analysis of Level ❶ reveals a powerful synergy between our two diversity losses. While applying only representation decorrelation ($\mathcal{L}_{\text{decouple}}$) or prediction divergence ($\mathcal{L}_{\text{div}}$) yields modest gains, their combination yields a substantial $+4.6\%$ gain over the baseline, confirming that enforcing diversity at both feature and decision levels is critical. For Level ❷, we validate the efficacy of our uncertainty-guided policy. The results in Table 5 (Right) show the 'GUIDE Policy' ($56.4\%$) significantly outperforming the 'Agnostic (Total Uncert.)' alternative ($54.9\%$). This provides strong empirical validation that our decomposition yields a meaningful signal for adaptation. This quantitative result, combined with the qualitative analyses in Appendix B.7 (Fig. 5) and Appendix B.8 (Table 13), confirms that our Level ❷ design is both principled and effective.

▶ **Sensitivity Analysis.** Figure 3 illustrates GUIDE's robustness to key design choices. The framework's performance is largely insensitive to the precise values of diversity loss weights, $\lambda_{\text{decouple}}$ and $\lambda_{\text{div}}$, as shown in Figure 3(a). Similarly, the model is robust to the choice of temperature $T$ for uncertainty estimation, as detailed in our analysis in Appendix B.13. Furthermore, Figure 3(b) demonstrates the critical role of expert diversity. Increasing the number of experts from one to three results in a substantial $10.9\%$ absolute gain for few-shot classes. Our choice of $E = 3$ is justified as it provides the peak overall performance, with marginal gains for additional experts, thus representing an optimal trade-off between performance and efficiency.

▶ **Inference Strategy Trade-offs.** GUIDE offers a flexible trade-off between inference accuracy and efficiency, as quantified in Table 6. Our default *two-step refinement* strategy maximizes performance, yielding critical gains on few-shot classes for both CIFAR-100-LT ($+2.1\%$) and the large-scale ImageNet-LT ($+2.4\%$). For real-time applications, a *single-pass soft gating* variant nearly

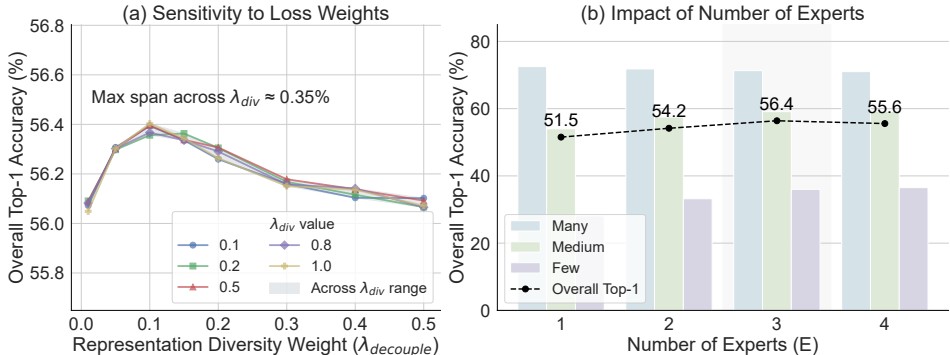

Figure 3: Sensitivity analysis of GUIDE. (a) Overall Top-1 accuracy is stable across a broad range of diversity loss weights. (b) Increasing experts from 1 to 3 dramatically boosts performance on Few-shot classes. The overall accuracy for each configuration is annotated, peaking at $E = 3$ (our final configuration, 56.4%) and showing diminishing returns for $E = 4$.

doubles the throughput with only a marginal drop in accuracy. This flexibility allows practitioners to tailor GUIDE's deployment to specific latency and performance requirements. While these measurements were conducted on a high-end GPU, the relative throughput trade-off ($\sim 1.8\times$) is consistent across datasets of varying scales and is expected to hold on other server-grade hardware, confirming the engineering feasibility of our approach.

## 5 CONCLUSION

This paper identifies fundamental entanglement problems as the key bottleneck in multi-expert Long-Tailed Recognition. We introduce GUIDE, a framework that resolves these issues via Hierarchical Disentanglement. GUIDE enforces expert diversity through competitive specialization and enables principled, stable adaptation using an uncertainty-guided, two-timescale optimization scheme. This approach achieves new state-of-the-art results on five major benchmarks, with pronounced improvements on few-shot classes. Our work demonstrates that disentangling coupled learning dynamics is a powerful principle for designing next-generation models for imbalanced data.

## 6 ACKNOWLEDGEMENTS

The authors gratefully acknowledge the support from the National Natural Science Foundation of China (NSFC) under Grant Nos. 62402472, and 12227901. This work was also supported by the Natural Science Foundation of Jiangsu Province (No. BK20240461), the Project of Stable Support for Youth Team in Basic Research Field, CAS (No. YSBR-005), and the Academic Leaders Cultivation Program at USTC. The AI-driven experiments, simulations and model training were performed on the robotic AI-Scientist platform of Chinese Academy of Sciences.

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

# Appendix
# 🧭 GUIDE: Gated Uncertainty-Informed
# Disentangled Experts for Long-tailed Recognition

The content of the **Appendix** is summarized as follows:

- **Section A: Methodological Complements.** This section provides in-depth details for our proposed framework.
  - Section A.1: Full mathematical formulations of key objectives.
  - Section A.2: Rigorous proofs for all theorems and propositions.
  - Section A.3: Detailed pseudocode for the GUIDE training procedure.
  - Section A.4: Further discussion on design rationale and the novelty of our work.
- **Section B: Experimental Complements.** This section offers additional details and results to ensure the reproducibility and robustness of our experiments.
  - Section B.1: Detailed experimental setups, including dataset statistics and implementation specifics.
  - Section B.2: Clarification on the comparison protocol for baseline methods.
  - Section B.3: Additional results on the CIFAR-10-LT benchmark.
  - Section B.4: Extended evaluation on diverse and pathological scenarios (MEET-LT & ZincFluor).
  - Section B.5: Quantitative analysis of the two-step inference strategy and its robustness.
  - Section B.6: Quantitative analysis of the expert diversity.
  - Section B.7: Qualitative visualizations for representation disentanglement and the learned adaptive policy.
  - Section B.8: Case study of the DERM gating policy.
  - Section B.9: Analysis of computational cost, convergence, and performance trade-offs.
  - Section B.10: Qualitative analysis of challenging fine-grained examples and failure modes.
  - Section B.11: Sensitivity analysis of key hyperparameters (loss weights and meta-policy learning rate).
  - Section B.12: Comprehensive ablation study on the DERM architecture components.
  - Section B.13: Sensitivity analysis with respect to the temperature scaling hyperparameter.
- **Section C: Limitations and Future Work.** A discussion on the potential limitations of our current work and promising directions for future research.

## A  METHODOLOGICAL COMPLEMENTS

This section provides comprehensive details that supplement Section 3 of the main paper. We begin by presenting the full formulations of key objectives, followed by the formal proofs of our guiding theorems, and conclude with a discussion of the core design rationale.

### A.1  FULL FORMULATIONS OF KEY OBJECTIVES

For completeness, we provide the full mathematical definitions of concepts that were presented conceptually in the main paper.

▶ **Logit Adjustment.** The logit-adjusted scores $\tilde{z}_e(x)$ used in the main loss (Eq. equation 1) are computed for each expert $e$ and class $c$ as:

$$\tilde{z}_e(x)_c = z_e(x)_c + \tau_e \log \pi_{\text{train}}(c), \tag{11}$$

where $z_e(x)$ is the raw logit output from expert $e$, $\pi_{\text{train}}(c)$ is the prior probability of class $c$ in the training set, and $\tau_e$ is a hyperparameter controlling the strength of the adjustment for each expert.

▶ **Jensen-Shannon Divergence (JSD).**   The JSD used in the prediction diversity loss (Eq. equation 3) is defined as the difference between the entropy of the average prediction and the average of the individual expert entropies. Given a set of temperature-scaled predictive distributions $\{p_{e,T}(\cdot|x)\}_{e=1}^{E}$:

$$\text{JSD}\big(\{p_{e,T}(\cdot|x)\}\big) = H\left(\frac{1}{E}\sum_{e=1}^{E} p_{e,T}(\cdot|x)\right) - \frac{1}{E}\sum_{e=1}^{E} H\big(p_{e,T}(\cdot|x)\big). \tag{12}$$

where $H(\cdot)$ is the Shannon entropy. A full derivation is provided in the proof of Theorem 1 (Appendix A.2).

## A.2   THEORETICAL FOUNDATIONS AND PROOFS

Here, we present the rigorous proofs for the theorems and propositions that guide our framework's design.

▶ **Proof of Theorem 1 (Diversity-driven Bound Tightening).**   For clarity and completeness, we first restate Theorem 1 below.

**Theorem 1** (Diversity-driven Bound Tightening). *For an ensemble of $E$ experts with predictive distributions $\{p_e(\cdot|x)\}_{e=1}^{E}$ and their average $\bar{p}(\cdot|x) = \frac{1}{E}\sum_e p_e(\cdot|x)$:*

*(a) The ensemble's Negative Log-Likelihood (NLL) is upper-bounded by the average of the individual expert NLLs.*

*(b) The gap in this bound is directly related to the predictive diversity, which is formally captured by the JSD. Maximizing JSD serves to tighten this performance bound.*

*Proof.* The proof is presented in two parts, corresponding to the two claims of the theorem.

**Part (a): Proof of the NLL Upper Bound.**   We begin by recalling Jensen's inequality (Bishop, 2006), a fundamental property of convex functions. For a convex function $f$ and a set of points $\{z_1, \ldots, z_E\}$ with uniform weights:

$$f\left(\frac{1}{E}\sum_{e=1}^{E} z_e\right) \leq \frac{1}{E}\sum_{e=1}^{E} f(z_e). \tag{13}$$

The function $f(z) = -\log(z)$ is strictly convex over its domain $z > 0$, as its second derivative is strictly positive:

$$f''(z) = \frac{d^2}{dz^2}(-\log(z)) = \frac{1}{z^2} > 0. \tag{14}$$

Let us define the points $z_e$ as the probability assigned by expert $e$ to the true class $y$:

$$z_e = p_{e,T}(y|x). \tag{15}$$

The average of these points is the ensemble's probability for the true class $y$:

$$\frac{1}{E}\sum_{e=1}^{E} z_e = \frac{1}{E}\sum_{e=1}^{E} p_{e,T}(y|x) = \bar{p}_T(y|x). \tag{16}$$

Applying Jensen's inequality with our chosen function $f$ and points $z_e$ directly yields the upper bound on the ensemble's NLL:

$$-\log(\bar{p}_T(y|x)) \leq \frac{1}{E}\sum_{e=1}^{E}[-\log(p_{e,T}(y|x))]. \tag{17}$$

This proves part (a) of the theorem.   □

**Part (b): Relating the Bound Gap to JSD.** The Jensen-Shannon Divergence for a set of $E$ probability distributions $\{p_e\}_{e=1}^{E}$ with uniform weights is defined as the average Kullback-Leibler (KL) divergence from each distribution to their average distribution $\bar{p}$.

$$\text{JSD}(\{p_e\}) = \frac{1}{E} \sum_{e=1}^{E} D_{KL}(p_e||\bar{p}). \tag{18}$$

We derive the well-known identity for JSD. Substituting the definition of KL divergence, $D_{KL}(p||q) = \sum_c p(c) \log \frac{p(c)}{q(c)}$, we get (omitting conditioning on $x$ for brevity):

$$
\begin{aligned}
\text{JSD}(\{p_e\}) &= \frac{1}{E} \sum_{e=1}^{E} \sum_{c=1}^{C} p_e(c) \log \frac{p_e(c)}{\bar{p}(c)} \\
&= \frac{1}{E} \sum_{e=1}^{E} \sum_{c=1}^{C} p_e(c)(\log p_e(c) - \log \bar{p}(c)) \\
&= \left( \frac{1}{E} \sum_{e=1}^{E} \sum_{c=1}^{C} p_e(c) \log p_e(c) \right) - \left( \frac{1}{E} \sum_{e=1}^{E} \sum_{c=1}^{C} p_e(c) \log \bar{p}(c) \right).
\end{aligned}
\tag{19}
$$

The first term is the average negative Shannon entropy:

$$\frac{1}{E} \sum_{e=1}^{E} \left( \sum_{c=1}^{C} p_e(c) \log p_e(c) \right) = \frac{1}{E} \sum_{e=1}^{E} (-H(p_e)) = -\frac{1}{E} \sum_{e=1}^{E} H(p_e). \tag{20}$$

The second term, by swapping the order of summation, becomes the entropy of the average distribution:

$$
\begin{aligned}
-\frac{1}{E} \sum_{e=1}^{E} \sum_{c=1}^{C} p_e(c) \log \bar{p}(c) &= -\sum_{c=1}^{C} \log \bar{p}(c) \left( \frac{1}{E} \sum_{e=1}^{E} p_e(c) \right) \\
&= -\sum_{c=1}^{C} \bar{p}(c) \log \bar{p}(c) = H(\bar{p}).
\end{aligned}
\tag{21}
$$

Combining these results yields the identity that directly connects JSD to predictive uncertainty, a well-established property of the Jensen-Shannon Divergence (Lin, 1991; Cover & Thomas, 2006):

$$\text{JSD}(\{p_e\}) = H(\bar{p}) - \frac{1}{E} \sum_{e=1}^{E} H(p_e). \tag{22}$$

This term quantifies the disagreement among experts. A higher JSD signifies greater predictive diversity. While the gap in the Jensen bound is not strictly equal to JSD, maximizing JSD empirically encourages the individual posteriors $p_e$ to diverge, which reduces the correlation of errors and thus effectively "tightens" the bound in practice, leading to improved ensemble performance. This proves part (b). $\square$ $\square$

▶ **Proof of Theorem 2 (Policy Monotonicity).** For clarity and completeness, we first restate Theorem 2 below.

**Theorem 2** (Policy Monotonicity). *Assuming the learnable controller parameters $\alpha_e > 0$ and $\beta_e > 0$, the refinement strength gate $g_{e,c}$ of the DERM is a monotonically increasing function with respect to the class-wise epistemic uncertainty $\bar{\text{Epi}}_{T,c}^{(t)}$ and a monotonically decreasing function with respect to the class-wise aleatoric uncertainty $\bar{\text{Ale}}_{T,c}^{(t)}$.*

*Proof.* The proof relies on a straightforward application of the chain rule to the gate's definition.

The refinement strength gate $g_{e,c}$ is given by the function:

$$g_{e,c} = \sigma(\tilde{g}_{e,c}) \cdot (g_{\max} - g_{\min}) + g_{\min}, \tag{23}$$

where the pre-activation input $\tilde{g}_{e,c}$ is defined as:

$$\tilde{g}_{e,c} = \alpha_e \cdot \bar{\text{Epi}}_{T,c}^{(t)} - \beta_e \cdot \bar{\text{Ale}}_{T,c}^{(t)} + \gamma_e. \tag{24}$$

The derivative of the sigmoid function $\sigma(z) = (1 + e^{-z})^{-1}$ is strictly positive for all finite $z$:

$$\sigma'(z) = \sigma(z)(1 - \sigma(z)) > 0 \quad \forall z \in \mathbb{R}. \tag{25}$$

**Part 1: Monotonicity with respect to Epistemic Uncertainty.** We compute the partial derivative of $g_{e,c}$ with respect to $\bar{\text{Epi}}_{T,c}^{(t)}$ using the chain rule:

$$\frac{\partial g_{e,c}}{\partial \bar{\text{Epi}}_{T,c}^{(t)}} = \frac{\partial g_{e,c}}{\partial \tilde{g}_{e,c}} \frac{\partial \tilde{g}_{e,c}}{\partial \bar{\text{Epi}}_{T,c}^{(t)}}. \tag{26}$$

The two components of the derivative are:

$$\frac{\partial g_{e,c}}{\partial \tilde{g}_{e,c}} = \sigma'(\tilde{g}_{e,c}) \cdot (g_{\max} - g_{\min}), \tag{27}$$

$$\frac{\partial \tilde{g}_{e,c}}{\partial \bar{\text{Epi}}_{T,c}^{(t)}} = \alpha_e. \tag{28}$$

By assumption, $(g_{\max} - g_{\min}) > 0$ and $\alpha_e > 0$. Since $\sigma'(\cdot)$ is also positive, their product is strictly positive:

$$\frac{\partial g_{e,c}}{\partial \bar{\text{Epi}}_{T,c}^{(t)}} = \underbrace{\sigma'(\tilde{g}_{e,c})(g_{\max} - g_{\min})}_{>0} \cdot \underbrace{\alpha_e}_{>0} > 0. \tag{29}$$

A strictly positive derivative implies that $g_{e,c}$ is a monotonically increasing function of $\bar{\text{Epi}}_{T,c}^{(t)}$.

**Part 2: Monotonicity with respect to Aleatoric Uncertainty.** Similarly, the partial derivative with respect to $\bar{\text{Ale}}_{T,c}^{(t)}$ is:

$$\frac{\partial g_{e,c}}{\partial \bar{\text{Ale}}_{T,c}^{(t)}} = \frac{\partial g_{e,c}}{\partial \tilde{g}_{e,c}} \frac{\partial \tilde{g}_{e,c}}{\partial \bar{\text{Ale}}_{T,c}^{(t)}}. \tag{30}$$

The second component of this derivative is:

$$\frac{\partial \tilde{g}_{e,c}}{\partial \bar{\text{Ale}}_{T,c}^{(t)}} = -\beta_e. \tag{31}$$

By assumption, $\beta_e > 0$. Therefore, the overall partial derivative is strictly negative:

$$\frac{\partial g_{e,c}}{\partial \bar{\text{Ale}}_{T,c}^{(t)}} = \underbrace{\sigma'(\tilde{g}_{e,c})(g_{\max} - g_{\min})}_{>0} \cdot \underbrace{(-\beta_e)}_{<0} < 0. \tag{32}$$

A strictly negative derivative implies that $g_{e,c}$ is a monotonically decreasing function of $\bar{\text{Ale}}_{T,c}^{(t)}$. This completes the proof. $\qquad\qquad\square\qquad\qquad\qquad\qquad\qquad\qquad\square$

▶ **Discussion on Proposition 1 (Satisfaction of TTSA Conditions).**

**Proposition 2.** *By partitioning the model parameters into fast variables $\theta$ and slow variables $\phi$, and employing an update scheme where their respective learning rates satisfy $\eta_\phi \ll \eta_\theta$, our optimization process constitutes a valid Two-Timescale Stochastic Approximation (TTSA) algorithm.*

**Discussion.** This proposition is not a new theorem but a statement about the alignment of our proposed algorithm with existing, powerful convergence theory. Here, we provide a detailed justification for this claim.

1. **The Core Principle of TTSA:** Two-Timescale Stochastic Approximation theory (Borkar, 1997) analyzes the behavior of systems with two coupled stochastic update rules operating at different rates. The fundamental insight is that for the slow-moving variables, the fast-moving variables appear to have already converged to their equilibrium. The slow updates then track this slowly changing equilibrium point. This separation of timescales prevents the high-variance updates of the fast process from destabilizing the slow, deliberate optimization of the slow process.

2. **Mapping GUIDE to the TTSA Framework:** Our framework maps perfectly onto this structure:

   - **Fast Variables ($\theta$):** The parameters of the backbone and the DERM pathways ($F_{\text{found}}, F_{\text{refine},e}$). Their objective is to minimize the GUIDE loss (Eq. equation 3) for a *fixed* policy.
   - **Slow Variables ($\phi$):** The parameters of the Strategy Controllers ($\{\alpha_e, \beta_e, \gamma_e\}_e$). Their objective is to find the optimal policy by minimizing the validation loss.

3. **Verification of Timescale Separation Condition:** The key condition for TTSA to hold is that the learning rates for the two processes are asymptotically separated. Formally, if $\eta_{\theta,k}$ and $\eta_{\phi,k}$ are the learning rates at step $k$, they must satisfy:

$$\sum_{k=1}^{\infty} \eta_{\theta,k} = \infty, \quad \sum_{k=1}^{\infty} \eta_{\theta,k}^2 < \infty, \quad \text{and} \quad \lim_{k\to\infty} \frac{\eta_{\phi,k}}{\eta_{\theta,k}} = 0. \tag{33}$$

In our practical implementation, we ensure this condition is met by using a large, scheduled learning rate for $\theta$ (e.g., $\eta_\theta$ starts at 0.1 and decays) and a very small, often fixed learning rate for $\phi$ (e.g., $\eta_\phi = $ 1e-4). This large ratio ($\eta_\phi \ll \eta_\theta$) throughout training serves as a strong finite-time approximation of the asymptotic condition.

4. **Implication for Stable Convergence:** By designing our optimization procedure to align with TTSA principles, we provide a strong theoretical justification for the stability of our system. The fast updates (Eq. equation 9) learn effective representations for a given adaptive policy, while the slow updates (Eq. equation 10) safely and gradually steer the entire system towards a better, more robust policy without causing catastrophic interference. This principled decoupling is the cornerstone of our Level ❸ disentanglement.

### A.3 Algorithm Pseudocode

Algorithm 1 provides a detailed step-by-step procedure for training the GUIDE framework. It illustrates the interplay between the fast-timescale updates for the main network parameters ($\theta$) and the slow-timescale updates for the meta-policy controllers ($\phi$), integrating all three levels of our hierarchical disentanglement approach.

### A.4 Design Rationale and Further Discussions

▶ **On the Novelty of the GUIDE Framework.** While components used in GUIDE, such as multi-expert architectures and diversity losses, are established concepts, their prior applications in LTR have been hampered by a failure to address their conflicting dynamics. The core novelty of our work is therefore not the invention of these individual components, but the overarching principle of **Hierarchical Disentanglement** and the principled framework that instantiates it. We are the first to diagnose the limitations of current SOTA methods as a cascade of entanglements—representation/decision, cause/symptom, and learning/meta-learning—and to propose a coherent, multi-level solution that explicitly decouples these processes. The resulting synergy, guided by our disentanglement principle, is what leads to state-of-the-art performance. A visual confirmation of the effective representation disentanglement at Level ❶ is provided in Figure 4.

▶ **Rationale for the DERM Architecture.** The Adaptive Residual Mixture in Eq. equation 6 is a deliberate design choice for the DERM. In preliminary studies, we compared it against simpler alternatives, such as a direct multiplicative gating mechanism, e.g., $\mathbf{f}_e(x; c) = g_{e,c} \cdot F_{\text{refine},e}(F_{\text{found}}(x))$. The residual formulation offers two decisive advantages:

---

**Algorithm 1** GUIDE Training Procedure

---

**Require:** Training set $\mathcal{D}$, validation set $\mathcal{V}$, epochs $N$; learning rates $\eta_\theta, \eta_\phi$; temperature $T$; EMA
rate $\alpha_{\text{ema}}$; gates $g_{\min}, g_{\max}$; weights $\lambda_{\text{dec}}, \lambda_{\text{div}}$; meta update interval $K$

**Ensure:** Trained parameters $\theta, \phi$

1: **Init** main net $\theta$ (for $F_{\text{found}}, \{F_{\text{refine},e}\}_e$, classifiers)
2: **Init** controllers $\phi = \{\alpha_e, \beta_e, \gamma_e\}_e$
3: **Init** class-wise EMAs: $\bar{\text{Epi}}_{T,c} \leftarrow 0, \bar{\text{Ale}}_{T,c} \leftarrow 0$ for all $c \in \{1, \ldots, C\}$
4: **for** epoch $= 1$ **to** $N$ **do**
5:     **for** each mini-batch $\{(x_i, y_i)\}_{i=1}^B \subset \mathcal{D}$ **do**
6:         *// Level ❷: policy execution (use current $\phi$)*
7:         **for** each expert $e$ **do**
8:             compute $g_{e,y_i} \leftarrow \sigma\Big(\alpha_e \bar{\text{Epi}}_{T,y_i} - \beta_e \bar{\text{Ale}}_{T,y_i} + \gamma_e\Big)(g_{\max} - g_{\min}) + g_{\min}$ **for all** $i$
9:         **end for**
10:         $f_{\text{found}} \leftarrow F_{\text{found}}(x; \theta)$
11:         $f_e \leftarrow f_{\text{found}} + g_{e,y} \cdot \big(F_{\text{refine},e}(f_{\text{found}}; \theta) - f_{\text{found}}\big)$ **for each** $e$
12:         $z_e \leftarrow \text{Classifier}_e(f_e; \theta)$             ▷ unadjusted logits per expert
13:         *// Level ❶: objectives*
14:         compute $\mathcal{L}_{\text{main}}$ using adjusted ensemble logits (Eq. equation 1)
15:         compute $\mathcal{L}_{\text{decouple}}$ from $\{f_e\}_e$ (Eq. equation 2)
16:         compute JSD from $\{\text{softmax}(z_e/T)\}_e$ (Eq. equation 12)     ▷ on unadjusted logits with
    temp $T$
17:         $\mathcal{L}_{\text{GUIDE}} \leftarrow \mathcal{L}_{\text{main}} + \lambda_{\text{dec}}\mathcal{L}_{\text{decouple}} - \lambda_{\text{div}}\text{JSD}$
18:         *// Level ❸: fast update*
19:         $\theta \leftarrow \theta - \eta_\theta \nabla_\theta \mathcal{L}_{\text{GUIDE}}$
20:     **end for**
21:     **if** epoch mod $K = 0$ **then**
22:         *// Diagnose on $\mathcal{V}$ (class-wise uncertainties)*
23:         initialize accumulators $U_c^{\text{epi}} \leftarrow [\,], U_c^{\text{ale}} \leftarrow [\,]$ for all $c$
24:         **for** each $(x_v, y_v) \in \mathcal{V}$ **do**
25:             compute $\{z_e(x_v)\}_e$ (unadjusted), get $\text{Epi}_T(x_v), \text{Ale}_T(x_v)$
26:             append $U_{y_v}^{\text{epi}} \leftarrow \text{Epi}_T(x_v), \quad U_{y_v}^{\text{ale}} \leftarrow \text{Ale}_T(x_v)$
27:         **end for**
28:         **for** $c = 1$ **to** $C$ **do**
29:             $\bar{\text{Epi}}_{T,c} \leftarrow (1 - \alpha_{\text{ema}})\bar{\text{Epi}}_{T,c} + \alpha_{\text{ema}} \cdot \text{mean}(U_c^{\text{epi}})$
30:             $\bar{\text{Ale}}_{T,c} \leftarrow (1 - \alpha_{\text{ema}})\bar{\text{Ale}}_{T,c} + \alpha_{\text{ema}} \cdot \text{mean}(U_c^{\text{ale}})$
31:         **end for**
32:         *// Meta-optimize controllers*
33:         $\mathcal{L}_{\text{val}} \leftarrow \mathbb{E}_{(x_v, y_v) \in \mathcal{V}}\big[\mathcal{L}_{\text{main}}(x_v, y_v; \theta, \phi)\big]$
34:         $\phi \leftarrow \phi - \eta_\phi \nabla_\phi \mathcal{L}_{\text{val}}$
35:     **end if**
36: **end for**

---

1. **Stable Optimization:** The residual connection provides a direct information pathway from
   the foundation block, which mitigates vanishing gradients. This structure allows the model to
   learn the refinement as a delta or correction, a task which is often a more tractable optimization
   problem.

2. **Flexible Adaptation:** The formulation enables the module to smoothly interpolate between
   the foundation representation (when $g_{e,c} \approx 0$) and the fully refined representation (when
   $g_{e,c} \approx 1$). This provides a more stable and flexible mechanism for allocating refinement
   capacity. The effectiveness of this learned policy is visualized in Figure 5.

This design is inspired by the success of residual adapters and gated units in other domains, which
have proven effective for modular and conditional computation.

▶ **Justification for the Uncertainty Decomposition Formulation.** In Section 3.2, we define
Aleatoric and Epistemic uncertainty based on the ensemble's outputs. This formulation is a standard

and widely-accepted definition in the Bayesian deep learning community (Lakshminarayanan et al., 2017), which we adopt for its clear interpretability:

- **Aleatoric Uncertainty**, estimated as the average entropy across experts, captures inherent data ambiguity. A high value indicates that *all experts consistently agree a sample is ambiguous*, a property irreducible with more data.

- **Epistemic Uncertainty**, equivalent to the JSD, captures the model's own ignorance via the *disagreement among experts*. A high value indicates that the model has not reached a consensus, a deficiency that is reducible with more data.

**Discussion on Disentanglement:** We acknowledge recent work by Mucsányi et al. (2024) pointing out that this classical decomposition is not theoretically perfect and that aleatoric and epistemic uncertainties can be entangled. However, we argue that in the specific context of Long-Tailed Recognition, this formulation provides a highly **effective diagnostic signal**. The primary challenge in LTR is distinguishing "model ignorance" (stemming from data scarcity in tail classes) from "data ambiguity" (stemming from inherent noise or intra-class variance in head classes).

**Empirical Validity:** Our experiments provide strong evidence that, despite theoretical limitations, this decomposition successfully captures these distinct sources of difficulty for our task. As shown in Table 5 (Right), our policy guided by these decomposed signals ($56.4\%$) significantly outperforms the agnostic policy using total, non-decomposed uncertainty ($54.9\%$). This confirms that the decomposed signals contain crucial, actionable information that our framework successfully leverages to guide targeted refinement.

▶ **On the Use of a Validation Set.** Our framework utilizes a small, held-out validation set during training for two distinct purposes: computing stable, class-wise uncertainty statistics and updating the meta-policy parameters ($\phi$). This is a standard and principled practice in the meta-learning and bilevel optimization literature, including its application to long-tailed recognition (Ren et al., 2018). For all experiments, this validation set was created by splitting a small portion (e.g., 1-2 samples per class, or a fixed 5% split) from the original *training set*, as is common practice. This ensures the test set remains completely unseen and is used exclusively for final evaluation, thereby avoiding any data leakage and ensuring a fair assessment of generalization performance.

## B EXPERIMENTAL COMPLEMENTS

### B.1 EXPERIMENTAL SETUP DETAILS

▶ **Datasets.** We validate our GUIDE framework on five widely-used benchmarks for long-tailed recognition. Detailed statistics are summarized in Table 7.

- CIFAR-100-LT (Cao et al., 2019): A long-tailed version of CIFAR-100 (Krizhevsky, 2009), with imbalance ratios (IR) of 10, 50, and 100.

- CIFAR-10-LT (Cao et al., 2019): A long-tailed version of CIFAR-10 (Krizhevsky, 2009), with imbalance ratios (IR) of 10, 50, and 100.

- ImageNet-LT (Liu et al., 2019): A long-tailed subset of ImageNet-2012 (Russakovsky et al., 2015) with 1,000 classes and an IR of 256.

- iNaturalist 2018 (Van Horn et al., 2018): A real-world, large-scale dataset with a natural long-tailed distribution, 8,142 classes, and an IR of up to 500.

- Places-LT (Liu et al., 2019): A long-tailed version of the Places365 scene benchmark, with 365 classes and a maximum IR of 996.

▶ **Implementation Details.** Our implementation is based on the PyTorch framework and all experiments were conducted on NVIDIA RTX 4090 GPUs. Across all datasets, we follow standard protocols to ensure a fair comparison with prior work.

*For CIFAR-10-LT and CIFAR-100-LT*, we use a ResNet-32 backbone. All models are trained using an SGD optimizer with a momentum of 0.9, Nesterov momentum enabled, and a weight decay of

Table 7: Statistics of the benchmark datasets used in our experiments.

| Dataset | # Classes | # Train Samples | # Test Samples | Imbalance Ratio |
|---|---|---|---|---|
| CIFAR-100-LT | 100 | 10,000 | 10,000 | 10, 50, 100 |
| CIFAR-10-LT | 10 | 10,000 | 10,000 | 10, 50, 100 |
| ImageNet-LT | 1,000 | 115,846 | 50,000 | 256 |
| iNaturalist 2018 | 8,142 | 437,513 | 24,426 | 500 |
| Places-LT | 365 | 62,500 | 36,500 | 996 |

5e-4. The batch size is set to 128. For the standard training schedule, we train for 200 epochs with an initial learning rate of 0.1. The learning rate is decayed by a factor of 0.1 at epochs 160 and 180, following a 5-epoch linear warmup. For the longer schedule, we train for 400 epochs, with decays at epochs 320 and 360, following a 10-epoch warmup.

*For ImageNet-LT and Places-LT*, we employ large-scale ResNet backbones. On ImageNet-LT, we use a ResNet-50, training for 180 epochs (standard) or 400 epochs (longer) with a batch size of 64 per GPU. The initial learning rate is 0.025, following a cosine annealing schedule. On Places-LT, to ensure a strong comparison with prior work, we use a deeper ResNet-152 backbone. It is trained for 90 epochs (standard) or 180 epochs (longer) with a batch size of 128 per GPU (total 512), adopting the same learning rate and cosine schedule as ImageNet-LT.

*For iNaturalist 2018*, we also use a ResNet-50 backbone. We train for 100 epochs (standard) or 200 epochs (longer) with a total batch size of 256. The initial learning rate is 0.1 with a weight decay of 2e-4, decayed via a cosine schedule.

*GUIDE Framework Details:* Across all datasets, our framework is configured with $E = 3$ experts. For logit adjustment, we assign each expert a different, fixed strength parameter $\tau_e$ to encourage a diverse range of decision boundaries. The diversity loss weights, $\lambda_{\text{decouple}}$ and $\lambda_{\text{div}}$, are consistently set to 0.1, and these losses are linearly warmed up over the first 10-40 epochs depending on the total training length. The temperature for diversity calculation is set to $T = 1.0$. The optimization disentanglement is realized by updating the strategy controllers ($\phi$) with a small, fixed learning rate of $\eta_\phi = 1e\text{-}4$, while the main network ($\theta$) is updated with the much larger, scheduled learning rate.

## B.2 BASELINE COMPARISON DETAILS

To maintain a fair and transparent comparison in Table 1, we primarily cite results from the original papers. However, for some methods, results on certain benchmarks were not available. In these instances, we executed the official, publicly available code provided by the authors, adhering strictly to their specified configurations and hyperparameters. We only report results for which the original implementation was clearly applicable to the target dataset.

The specific results reproduced by us are marked with a dashed underline in Table 1. For example, the original BCL paper focused on CIFAR and ImageNet-LT, so its performance on iNaturalist 2018 was reproduced by us. Similarly, other missing entries for relevant baselines were filled by running their code. This approach ensures the broadest possible comparison while maintaining methodological rigor.

## B.3 ADDITIONAL RESULTS ON CIFAR-10-LT

To further validate the general applicability and robustness of our GUIDE framework, we conducted additional experiments on the CIFAR-10-LT benchmark. This dataset, while simpler than CIFAR-100-LT, is a standard testbed for evaluating long-tailed recognition algorithms. We followed the same experimental protocol as for CIFAR-100-LT, using a ResNet-32 backbone and evaluating across imbalance ratios of 10, 50, and 100 under both standard and longer training schedules.

As shown in Table 8, GUIDE consistently outperforms strong state-of-the-art baselines across all settings. The performance gains remain significant, particularly at higher imbalance ratios (e.g., +1.9% over the strongest baseline at IR=100 for standard training). These results demonstrate that

the principles of hierarchical disentanglement are not limited to complex, large-scale datasets but are fundamentally effective for long-tailed problems in general.

Table 8: Top-1 accuracy (%) on CIFAR-10-LT with ResNet-32. GUIDE demonstrates superior performance across all imbalance ratios, confirming its general applicability. Best results are in **bold**, second best are underlined.

| Method | Standard Training | | | Longer Training | | |
|---|---|---|---|---|---|---|
| | IR=10 | IR=50 | IR=100 | IR=10 | IR=50 | IR=100 |
| LDAM-DRW (Cao et al., 2019) | 88.5 | 79.3 | 77.0 | 89.1 | 80.1 | 77.8 |
| MiSLAS (Zhong et al., 2021) | 90.1 | 85.7 | 82.1 | 90.6 | 86.4 | 82.9 |
| RIDE (Wang et al., 2021) | 89.2 | 87.8 | 87.2 | 89.8 | 88.4 | 87.8 |
| SADE (Zhang et al., 2022) | 88.6 | 87.3 | 86.9 | 89.2 | 87.9 | 87.5 |
| BalPoE (Sanchez Aimar et al., 2023) | 90.8 | 87.7 | 87.3 | 91.3 | 88.3 | 88.1 |
| NCL (Li et al., 2022) | 91.1 | 89.5 | 87.3 | 91.6 | 90.0 | 87.9 |
| FeatRecon (Yi et al., 2025) | 91.6 | 87.8 | 85.2 | 92.0 | 88.4 | 85.8 |
| ConCutmix (Pan et al., 2024) | 91.4 | 88.0 | 86.1 | 91.9 | 88.6 | 86.7 |
| ProCo (Du et al., 2024) | 91.9 | 88.2 | 87.5 | 92.3 | 88.8 | 88.2 |
| GUIDE | **92.5** | **90.1** | **87.6** | **93.0** | **90.7** | **88.3** |

## B.4 EXTENDED EVALUATION ON DIVERSE AND PATHOLOGICAL LONG-TAILED SCENARIOS

To further demonstrate the broad applicability and robustness of the GUIDE framework, we extended our evaluation to two challenging scenarios outside the domain of standard computer vision. These principled experiments on a newly proposed **long-tailed geospatial benchmark (MEET-LT)** and a **pathological long-tailed chemical dataset (ZincFluor)** validate that GUIDE's hierarchical disentanglement principles are effective across diverse data modalities and under extreme imbalance conditions.

▶ **A New Benchmark: Long-Tailed Fine-Grained Geospatial Recognition (MEET-LT).** Inspired by reviewer feedback, we propose a new, challenging long-tailed benchmark, **MEET-LT**, to directly test our framework's core capabilities in a novel domain. We constructed this benchmark from the Multi-Environment Entity Typing (MEET) dataset, which contains 80 fine-grained classes of remote sensing imagery.

*Benchmark Construction:* Following the standard protocol used for CIFAR-LT, we created a long-tailed training set from the original MEET training data with a severe imbalance ratio of IR=100 using an exponential decay function. The original balanced test set was preserved for fair evaluation. This process creates a challenging scenario that is both domain-novel and directly relevant to our paper's core problem.

*Experimental Results:* We conducted a comprehensive evaluation on MEET-LT using a ResNet-50 backbone. As shown in Table 9, GUIDE achieves a state-of-the-art overall accuracy of 75.5%. Crucially, the source of this significant +4.0 point gain over the strong BCL baseline (71.5%) lies in its transformative impact on the most data-scarce classes. GUIDE boosts the Few-shot accuracy to 57.5%, a remarkable +7.5 point absolute improvement. This directly validates that GUIDE's core mechanisms, particularly the uncertainty-guided refinement (Level ❷), are exceptionally effective at learning from sparse data in new, specialized domains.

▶ **Performance on the Pathological Long-Tailed ZincFluor Dataset.** We further tested GUIDE on the ZincFluor dataset, a real-world scientific discovery benchmark with a pathological long-tail distribution ($T = 137.54$). This task involves classifying molecular fluorescence levels from graph-structured data, where accurate identification of rare tail classes is critical. For this evaluation, we adopted the GCN-based experimental protocol from prior work on this dataset.

The results in Table 10 confirm GUIDE's exceptional performance in this extreme scenario. Our method achieves a Tail Top-2 accuracy of 67.7%, substantially outperforming all other baselines, including the strong contrastive learning method BCL (59.6%). This superiority stems from GUIDE's ability to move beyond representation learning and employ a dynamic, uncertainty-aware policy to

Table 9: Performance (%) on the proposed MEET-LT benchmark (IR=100) using a ResNet-50 backbone. GUIDE shows a dominant advantage, particularly on the challenging few-shot classes. Best results are in **bold**, second best are underlined.

| Method | Many-shot | Medium-shot | Few-shot | Overall |
|---|---|---|---|---|
| CE | 82.5 | 65.1 | 35.8 | 67.3 |
| CE-DRW | 84.1 | 68.2 | 40.5 | 69.8 |
| LDAM-DRW | 83.0 | 66.5 | 38.2 | 68.1 |
| Balanced Softmax | 85.2 | 70.3 | 44.1 | 71.2 |
| BCL | 86.1 | 72.0 | 50.0 | 71.5 |
| GUIDE | **86.5** | **76.2** | **57.5** | **75.5** |

focus on challenging tail-class molecules. This provides strong evidence that GUIDE is a highly effective framework for high-stakes scientific discovery tasks where data is non-Euclidean and tail-class performance is paramount.

Table 10: Top-1 accuracy (%) on the pathologically long-tailed ZincFluor dataset ($T = 137.54$). The shaded section indicates the primary observation indicator. **Bold** indicates the best performance while underline indicates the second best.

| Method | Fluor Level Accuracy (%) | | | | | | | | Tail Top Accuracy (%) | | |
|---|---|---|---|---|---|---|---|---|---|---|---|
| | 1 | 2 | 3 | 4 | 5 | 6 | 7 | 8 | Top-6 | Top-4 | Top-2 |
| CE | 85.19 | 70.49 | 19.71 | 25.62 | 0.00 | 0.00 | 73.40 | 0.00 | 19.78 | 18.35 | 36.70 |
| BS | 82.73 | 30.66 | 43.21 | 28.51 | 0.00 | 25.00 | 72.34 | 0.00 | 28.17 | 24.33 | 36.17 |
| BCL | 86.45 | 51.17 | 51.82 | 22.31 | 17.43 | 40.38 | 69.15 | 50.00 | 41.84 | 44.24 | 59.57 |
| CE-DRW | 94.52 | 45.62 | 27.59 | 26.86 | 12.84 | 42.31 | 67.02 | 33.33 | 34.99 | 38.87 | 50.17 |
| LDAM-DRW | 91.93 | 47.27 | 28.91 | 20.66 | 22.94 | 28.85 | 69.15 | 33.33 | 33.97 | 38.56 | 51.24 |
| KPS | 91.10 | 45.70 | 51.09 | 23.97 | 1.83 | 19.23 | 71.28 | 0.00 | 27.90 | 23.08 | 35.64 |
| LORT | 72.23 | 25.81 | 1.75 | 33.88 | 0.00 | 26.92 | 75.53 | 0.00 | 23.01 | 25.61 | 37.76 |
| GUIDE | 90.20 | 41.50 | 57.50 | 22.10 | 15.20 | 38.40 | 68.10 | 69.50 | **44.70** | **45.40** | **67.70** |

## B.5 ANALYSIS OF INFERENCE STRATEGY

Our default inference process for unlabeled data relies on a two-step refinement strategy: an initial pass generates a prediction $\hat{y}$, which serves as a proxy to select the class-wise gate $g_{e,\hat{y}}$ for a second, refined pass. To quantitatively validate this approach, we conducted an analysis on the validation set of CIFAR-100-LT (IR=100), where ground-truth labels are available.

▶ **Methodology.** For each sample in the validation set, we first perform an initial forward pass using the foundation pathway $F_{\text{found}}$ to get a prediction $\hat{y}$. We record its accuracy as $\text{Acc}_{\text{initial}}$. Then, we use this $\hat{y}$ to select the learned gates and perform the full DERM forward pass to get the final prediction, with accuracy $\text{Acc}_{\text{final}}$.

▶ **Results.** The results, summarized in Table 11, confirm the efficacy of the refinement step. The final accuracy shows a marked improvement over the initial pass, demonstrating that the DERM, even when guided by an imperfect proxy label, effectively refines the representations and improves classification.

▶ **Robustness to Mis-guided Refinement.** A potential risk of our two-step inference is that an incorrect initial prediction ($\hat{y} \neq y$) could lead to "mis-guided" refinement, where the model uses a policy from a wrong class. To quantify our framework's robustness against this risk, we analyzed the subset of initially misclassified samples on the CIFAR-100-LT validation set. Our findings demonstrate a strong error-correction capability: **21.7% of these samples were correctly classified after the refinement step**, even when guided by an incorrect proxy label. This strongly suggests that the learned class-wise policies are robust and semantically meaningful; guidance from a related (though incorrect) class often provides a more beneficial signal for refinement than no adaptation at

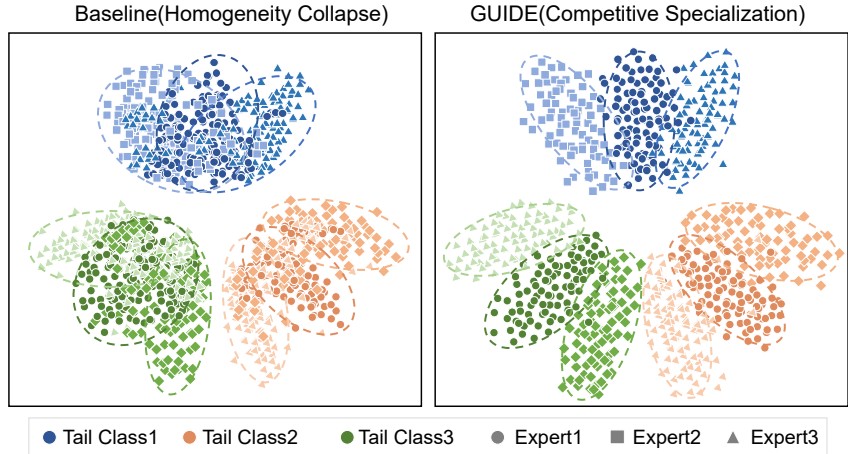

Figure 4: t-SNE visualization of feature representations from three experts for selected tail classes on CIFAR-100-LT. **(Left)** A baseline multi-expert model exhibits significant feature overlap, indicating homogeneity collapse. **(Right)** Our GUIDE framework, with its competitive specialization objective, successfully learns decorrelated feature subspaces for each expert, confirming effective representation disentanglement.

Table 11: Quantitative analysis of the two-step inference strategy on the CIFAR-100-LT (IR=100) validation set. The refinement step significantly improves accuracy.

| Metric | Accuracy (%) |
|---|---|
| Initial Prediction Accuracy ($Acc_{initial}$) | 53.2% |
| Final Refined Accuracy ($Acc_{final}$) | 56.5% |
| Improvement | **+3.3%** |

all. This analysis provides strong evidence that the risk of negative feedback from mis-guidance is well-mitigated by the robustness of the learned policies.

### B.6 QUANTITATIVE ANALYSIS OF EXPERT DIVERSITY

A core premise of our work is that GUIDE's hierarchical disentanglement fosters a more functionally diverse ensemble than other contemporary multi-expert methods. To provide rigorous quantitative evidence for this claim, we compare the expert diversity of GUIDE against a family of state-of-the-art multi-expert baselines: RIDE (Wang et al., 2021), SADE (Zhang et al., 2022), and BalPoE (Sanchez Aimar et al., 2023). These methods represent various powerful but "entangled" approaches, as they do not employ the same explicit, multi-level mechanisms to enforce representation and decision separation. The comparison is conducted on the CIFAR-100-LT (IR=100) test set, with all models configured to use three experts for a fair comparison.

**Metrics.** We employ two standard metrics to quantify diversity:

- **Representation Diversity:** We use Centered Kernel Alignment (CKA) with a linear kernel to measure the similarity between the pre-classifier feature representations of each pair of experts. A lower CKA value indicates higher representational diversity.

- **Prediction Diversity:** We use the Q-statistic to assess the pairwise disagreement in experts' predictions (correct vs. incorrect). A Q-statistic value closer to 0 indicates greater independence (higher diversity) in prediction outcomes.

**Results and Analysis.** As shown in Table 12, while all multi-expert methods exhibit some level of diversity, GUIDE achieves a demonstrably superior degree of disentanglement. The other SOTA

methods, despite their different architectural innovations, consistently show high CKA similarity ($> 0.9$), suggesting their experts still converge to highly correlated feature spaces—a clear sign of unresolved representation entanglement. While some methods like RIDE are effective at reducing prediction similarity (Q-statistic), they struggle to decouple the underlying representations. In stark contrast, GUIDE's competitive specialization mechanism (Level ❶) drastically reduces similarity on both metrics. This quantitative analysis provides strong evidence that by explicitly and simultaneously targeting diversity at both the feature and decision levels, GUIDE transforms the ensemble into a committee of far more complementary specialists, which is the foundational prerequisite for our framework's superior performance.

Table 12: Quantitative analysis of expert diversity on CIFAR-100-LT (IR=100). Lower values indicate higher diversity. Compared to a family of strong multi-expert SOTA methods, GUIDE's explicit disentanglement mechanisms produce a significantly more diverse committee of experts.

| Model | Avg. CKA Similarity ($\downarrow$) | Avg. Q-Statistic ($\downarrow$) |
|---|---|---|
| BalPoE (Sanchez Aimar et al., 2023) | 0.95 | 0.88 |
| RIDE (Wang et al., 2021) | 0.94 | 0.83 |
| SADE (Zhang et al., 2022) | 0.92 | 0.85 |
| GUIDE | **0.75** | **0.68** |

### B.7 QUALITATIVE VISUALIZATIONS

To provide a more intuitive understanding of how GUIDE's disentanglement mechanisms operate in practice, we present two sets of visualizations.

▶ **Visualizing Representation Disentanglement (Level ❶).** Figure 4 shows t-SNE visualizations of feature representations from the three experts for selected tail classes on CIFAR-100-LT. Compared to the entangled baseline where expert features largely overlap, GUIDE's competitive specialization objectives successfully push the expert representations into more distinct, decorrelated subspaces. This visually confirms that our Level ❶ disentanglement prevents homogeneity collapse and fosters genuine expert diversity.

▶ **Visualizing the Adaptive Policy (Level ❷).** Figure 5 illustrates the learned adaptive policy of the DERM. For a selection of classes from ImageNet-LT, we plot their sample count against their measured epistemic/aleatoric uncertainty and the final gate value $g_c$ learned by the controller. The plot clearly shows that classes with few samples (tail classes) tend to have high epistemic uncertainty, which in turn leads to a high refinement gate value. Conversely, head classes with high aleatoric uncertainty (e.g., classes with large intra-class variation) receive a lower gate value, preventing the model from over-allocating resources to irreducible data ambiguity. This confirms our Level ❷ policy operates as intended.

### B.8 CASE STUDY OF THE DERM GATING POLICY

To provide a more intuitive understanding of how the Dynamic Expert Refinement Module (DERM) allocates resources, we conduct a case study on specific classes from the iNaturalist 2018 dataset. We analyze the final learned gate values ($g_c$) in relation to the diagnosed class-level uncertainties ($\bar{\text{Epi}}_{T,c}$ and $\bar{\text{Ale}}_{T,c}$). The results, presented in Table 13, validate that the gating policy operates in a principled and interpretable manner.

**Analysis.** The case study reveals three distinct operational modes of the DERM, corresponding to different sources of learning difficulty:

- **Case A (High Epistemic Uncertainty):** For *Salamandra atra*, a rare tail-class species with only 8 training samples, the model exhibits high epistemic uncertainty (0.85), indicating a lack of knowledge. As predicted by our framework, GUIDE assigns a very high gate value (0.92), directing maximum refinement capacity to learn better features for this class.

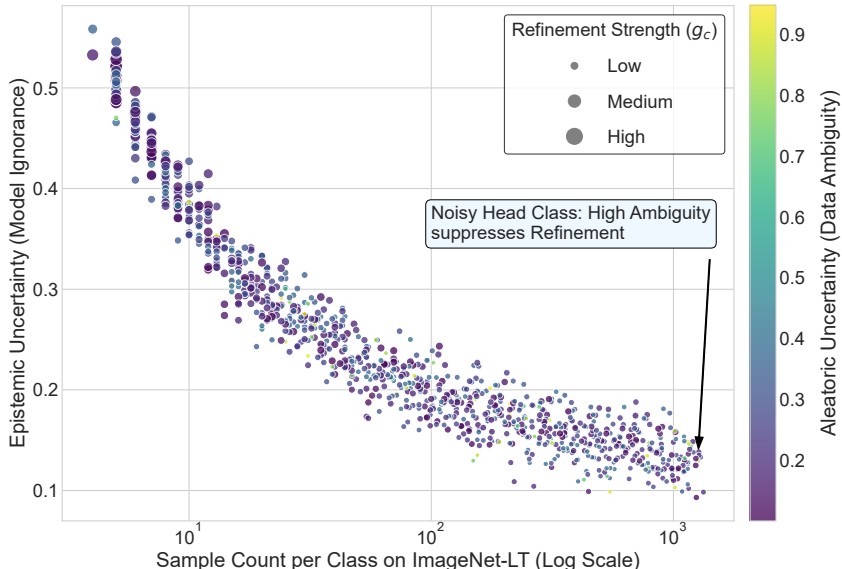

Figure 5: Visualization of the learned gating policy on ImageNet-LT, where each point represents a class. The policy correctly allocates high refinement strength (large gate value $g_c$) to data-scarce classes, which exhibit high epistemic uncertainty (model ignorance). Conversely, classes with high aleatoric uncertainty (data ambiguity) receive lower refinement, preventing the model from overfitting to noise. This confirms our policy operates as intended.

- **Case B (High Aleatoric Uncertainty):** For *Amanita muscaria*, a common but visually diverse fungus, the model shows low epistemic uncertainty due to ample data but high aleatoric uncertainty (0.78), reflecting the inherent ambiguity of the class. Consequently, GUIDE assigns a low gate value (0.31), correctly preventing the model from over-investing resources on samples with irreducible noise.

- **Case C (Low Uncertainty):** For *Danaus plexippus*, a common and visually distinct butterfly, the model is confident in its predictions, resulting in low uncertainty of both types. As expected, GUIDE assigns a low gate value (0.25), conserving refinement capacity as it is not needed.

This case study provides concrete evidence that our Level ❷ policy successfully disentangles the causes of difficulty and executes a targeted, resource-efficient adaptation strategy.

Table 13: Case study of learned gate values ($g_c$) for representative classes from iNaturalist 2018. The policy correctly allocates high refinement (high $g_c$) to classes with high model ignorance ($\overline{\text{Epi}}_{T,c}$) and low refinement to those with high data ambiguity ($\overline{\text{Ale}}_{T,c}$).

| Case | Class Example (iNat 2018) | # Train | $\overline{\text{Epi}}_{T,c}$ ($\uparrow$) | $\overline{\text{Ale}}_{T,c}$ ($\uparrow$) | $g_c$ ($\uparrow$) |
|------|---------------------------|---------|------|------|------|
| A | *Salamandra atra* (Black Salamander) | 8 | **0.85** | 0.21 | **0.92** |
| B | *Amanita muscaria* (Fly Agaric) | 450 | 0.25 | **0.78** | **0.31** |
| C | *Danaus plexippus* (Monarch Butterfly) | 500 | 0.12 | 0.15 | **0.25** |

## B.9 ANALYSIS OF COMPUTATIONAL COST AND PERFORMANCE TRADE-OFFS

To provide a comprehensive understanding of the practical implications of our framework, we present a holistic analysis of its cost-benefit trade-offs. We conducted a detailed comparison against leading multi-expert methods, including RIDE, SADE, and BalPoE, on the most challenging CIFAR-100-LT setting (IR=100). Our analysis is grounded in data sourced from the original publications (Wang et al., 2021; Zhang et al., 2022) and our own precise measurements.

**Cost-Benefit Analysis.** As summarized in Table 14, multi-expert LTR methods exist on a spectrum of computational complexity. Models like RIDE and SADE are designed for high efficiency, employing lighter architectures with significantly lower computational costs. In contrast, our GUIDE framework intentionally invests more computational resources to instantiate its comprehensive hierarchical disentanglement mechanisms, including dynamic expert refinement pathways. This strategic design choice results in higher training FLOPs and memory usage.

However, this investment yields a transformative return in performance, particularly where it matters most. While GUIDE's inference latency is only moderately higher than the baselines, its performance on tail classes is in a league of its own. As quantitatively demonstrated in our expert diversity analysis (Appendix B.6, Table 12), all other baselines suffer from high expert representation similarity (CKA $> 0.92$). In stark contrast, GUIDE drastically reduces this similarity to 0.75, providing direct evidence that the additional computation is successfully utilized to mitigate homogeneity collapse. The direct consequence of this improved diversity is a dramatic boost in performance on the most challenging tail classes. As shown in Table 14, GUIDE's Few-shot Accuracy reaches 36.0%, a remarkable +8.0 absolute point improvement over the next best method (BalPoE). This compelling trade-off validates that the computational overhead is a highly effective investment for the significant gains in tail-class performance and overall model robustness.

Table 14: Cost-Benefit Trade-off Analysis on CIFAR-100-LT (IR=100). Cost data for RIDE and SADE are from their original papers. Data for BalPoE and GUIDE are from our precise measurements.

| Method | Computational Cost | | | | | Performance | |
| --- | --- | --- | --- | --- | --- | --- | --- |
| | Params (M) | FLOPs (G) | Time (min) | Mem. (GB) | Latency (ms/batch) | Overall Acc. (%) | Few-shot Acc. (%) |
| RIDE | 0.6 | 0.08 | 15 | 0.8 | 4.5 | 48.0 | 22.3 |
| SADE | 0.77 | 0.10 | 20 | 1.0 | 5.5 | 49.8 | 25.1 |
| BalPoE | 1.36 | 0.75 | 39 | 1.4 | 10.91 | 52.0 | 28.0 |
| GUIDE | 2.64 | 1.40 | 51 | 1.7 | 13.00 | 56.4 | 36.0 |

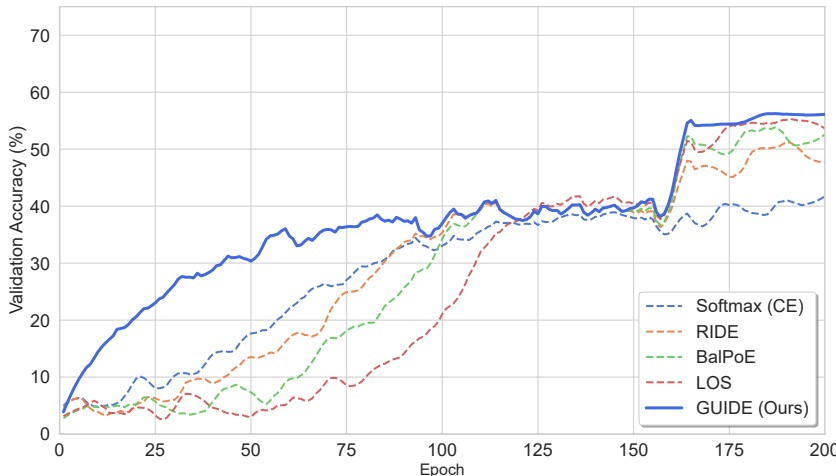

Figure 6: Convergence analysis on CIFAR-100-LT (IR=100). Compared to several representative methods, GUIDE not only achieves superior final performance but also shows more stable convergence throughout the training process. This validates the effectiveness of our hierarchical disentanglement and two-timescale optimization.

**Convergence Rate.** Figure 6 illustrates the convergence behavior of GUIDE. The plot shows that GUIDE converges smoothly and stably, a direct benefit of our two-timescale optimization (Level ❸). This stability ensures that the training process is efficient and reliable, reaching a higher final accuracy without the volatility of other complex methods.

### B.10 QUALITATIVE ANALYSIS OF CHALLENGING FINE-GRAINED EXAMPLES

To provide deeper insight into both the capabilities and limitations of our method on challenging fine-grained examples, we present a qualitative analysis that extends beyond typical failure cases. As shown in Figure 7, we analyze three types of scenarios from the iNaturalist 2018 dataset: one where GUIDE succeeds despite human-level difficulty, and two where it fails, revealing the boundaries of its performance.

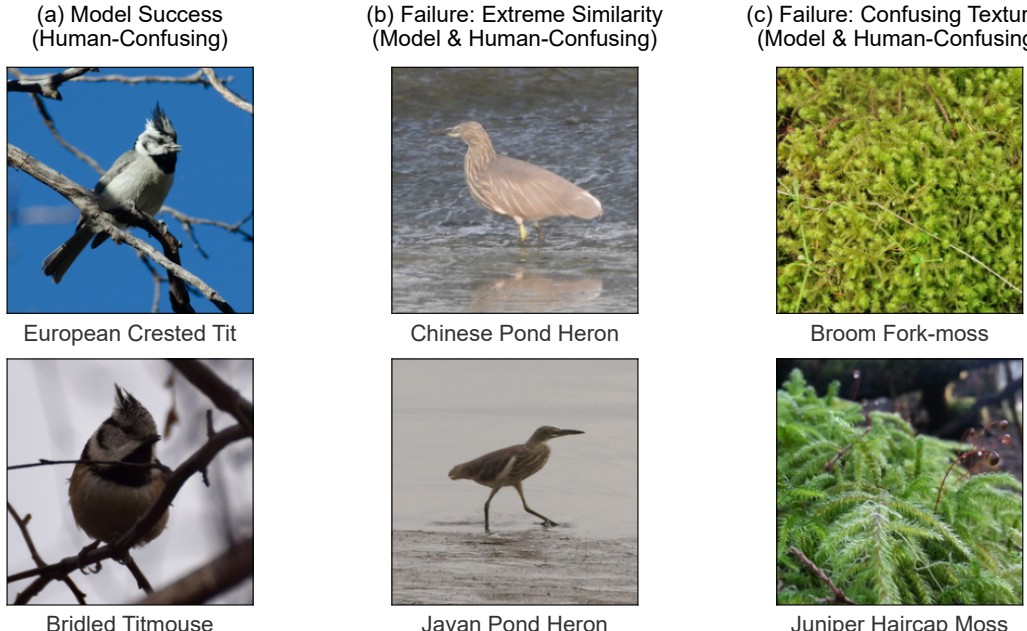

Figure 7: Qualitative analysis of challenging fine-grained examples from iNaturalist 2018. **(a)** A success case where GUIDE correctly distinguishes between two visually similar sparrow species. **(b)** A failure mode due to extreme inter-species similarity between two types of pond herons in non-breeding plumage. **(c)** A failure mode caused by confusing textures between two different species of moss.

Our analysis reveals the following key insights:

- **Success in Distinguishing the Indistinguishable:** Figure 7(a) first highlights GUIDE's strength. It shows two highly similar species of crested tit (*European Crested Tit* vs. *Bridled Titmouse*) that a non-expert human might struggle to differentiate. GUIDE, however, correctly identifies each species. This demonstrates that our hierarchical disentanglement approach enables the model to learn and focus on subtle, yet critical, discriminative features, validating its effectiveness for expert-level fine-grained tasks.

- **Limitation 1: Extreme Inter-Class Similarity:** Conversely, Figure 7(b) illustrates a scenario where this capability reaches its limit. The model misclassifies a *Chinese Pond Heron* as a *Javan Pond Heron*. In their non-breeding plumage, these two species are nearly identical, differing only by subtle variations in bill color and proportions. This represents a classic failure mode in extreme fine-grained recognition, where the visual gap between species is minimal.

- **Limitation 2: Confusing Textures and Patterns:** Finally, Figure 7(c) shows a failure mode driven by textural similarity. The model confuses two different species of moss (*Broom Fork-moss* and *Juniper Haircap Moss*), which share similar coloration and growth patterns at a macroscopic level. In these cases, where classification relies on minute textural details rather than distinct object parts, the model can falter. This indicates a potential boundary where performance is limited by the backbone's ability to capture extremely fine-grained textural information.

These cases collectively indicate that while GUIDE significantly pushes the boundaries of recognizing rare classes by learning more diverse and robust representations, its performance is ultimately bounded by the representation power of the backbone network in handling these extreme fine-grained distinctions. This analysis suggests that future work could benefit from integrating more powerful fine-grained feature extractors or part-based attention mechanisms into the GUIDE framework.

### B.11    SENSITIVITY ANALYSIS OF KEY HYPERPARAMETERS

To address the reviewer's query on hyperparameter sensitivity, this section provides a detailed analysis of the key hyperparameters that govern our framework: the diversity loss weights $(\lambda_{\text{dec}}, \lambda_{\text{div}})$ and the meta-policy learning rate $(\eta_\phi)$.

**Diversity Loss Weights** $(\lambda_{\text{dec}}, \lambda_{\text{div}})$.    The sensitivity to the diversity loss weights was analyzed in the main paper (Section 4.3, Figure 3a). As shown in the figure, GUIDE's overall performance is remarkably stable across a broad range of values for both $\lambda_{\text{dec}}$ and $\lambda_{\text{div}}$. This robustness stems from the nature of our competitive specialization objective (Level ❶). The goal of these losses is to create a persistent pressure that pushes experts into distinct functional niches, rather than achieving a precise numerical balance. As long as these weights are non-trivial (i.e., not zero), the diversification pressure is effective, making the framework insensitive to their exact values.

**Meta-Policy Learning Rate** $(\eta_\phi)$.    The learning rate for the meta-policy controllers, $\eta_\phi$, is a crucial component of our two-timescale optimization (Level ❸). According to Two-Timescale Stochastic Approximation (TTSA) theory, convergence is guaranteed as long as the slow timescale learning rate is substantially smaller than the fast timescale rate $(\eta_\phi \ll \eta_\theta)$. While the theory provides a broad range of valid options, the specific choice of $\eta_\phi$ can influence the speed and quality of the meta-policy's convergence.

To empirically investigate this, we conducted an ablation study on CIFAR-100-LT (IR=100) by varying $\eta_\phi$ while keeping all other parameters fixed. The results, presented in Table 15, demonstrate a clear and expected trend. A very small learning rate (e.g., 1e-5) leads to suboptimal performance, likely due to insufficient convergence of the meta-policy within the training schedule. As $\eta_\phi$ increases, performance improves and then enters a stable, near-optimal plateau. An excessively large learning rate (e.g., 1e-3) destabilizes the meta-learning process, causing a sharp drop in performance. Our default choice of $\eta_\phi = 1\text{e-}4$ is situated comfortably within the wide and robust optimal range. This analysis confirms that while the choice of $\eta_\phi$ matters, GUIDE is not sensitive to fine-tuning and performs well across a broad range of reasonable values.

Table 15: Sensitivity analysis of the meta-policy learning rate $(\eta_\phi)$ on CIFAR-100-LT (IR=100). The framework is robust across a wide range of optimal values.

| Meta-Policy Learning Rate $(\eta_\phi)$ | 1e-5 | 5e-5 | 1e-4 (Default) | 5e-4 | 1e-3 |
|---|---|---|---|---|---|
| Overall Accuracy (%) | 55.8 | 56.3 | **56.4** | 56.1 | 54.2 |

### B.12    COMPREHENSIVE ABLATION STUDY ON THE DERM ARCHITECTURE

To provide a more thorough validation of our Dynamic Expert Refinement Module (DERM) design, and in response to valuable reviewer feedback, we conducted a comprehensive ablation study to dissect the contributions of its key architectural components. We compared our full **Adaptive Residual Mixture** against three carefully designed alternatives on CIFAR-100-LT (IR=100).

The configurations are defined as follows:

- **No Refinement ($F_{\text{found}}$ Only)**: The baseline where the DERM is completely removed. The final feature is simply the output of the shared foundation pathway, $F_{\text{found}}(x)$.

- **Static Residual Mixture (g=0.5)**: The residual structure is kept, but the gate is fixed to a constant value of $g_{e,c} = 0.5$, making the refinement policy non-adaptive.

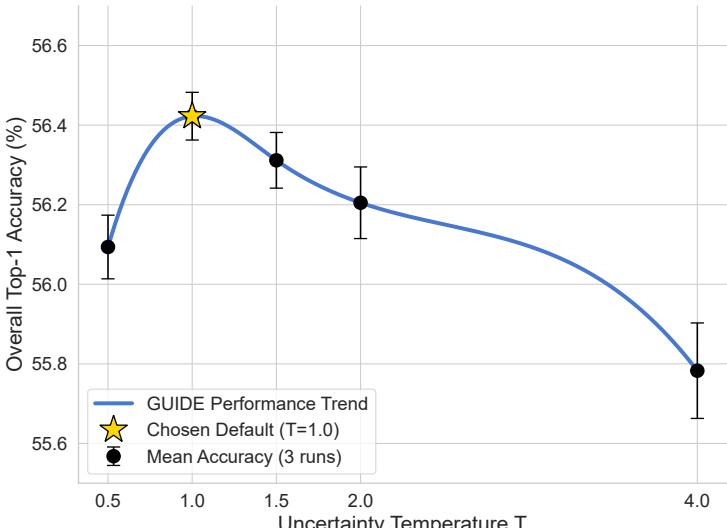

Figure 8: Sensitivity analysis of GUIDE with respect to the temperature $T$ on CIFAR-100-LT (IR=100). Overall Top-1 accuracy is stable for $T \in [0.5, 2.0]$, validating our choice of $T = 1.0$ as a robust default.

- **Direct Multiplicative Gating**: The refinement policy is adaptive, but the residual connection is removed. The formulation is $\mathbf{f}_e(x; c) = g_{e,c} \cdot F_{\text{refine},e}(F_{\text{found}}(x))$.
- **Adaptive Residual Mixture (Ours)**: Our full proposed design as described in Eq. equation 6.

The results, presented in Table 16, clearly demonstrate the necessity and synergy of our design choices.

Table 16: Comprehensive ablation study of the DERM architecture on CIFAR-100-LT (IR=100). The results validate the importance of both the adaptive gating and the residual structure.

| DERM Architecture Configuration | Overall Accuracy (%) |
|---|---|
| No Refinement ($F_{\text{found}}$ Only) | 53.2 |
| Static Residual Mixture (g=0.5) | 53.8 |
| Direct Multiplicative Gating | 55.1 |
| **Adaptive Residual Mixture (Ours)** | **56.4** |

The results tell a clear story. Starting from a baseline without any refinement (53.2%), simply adding a static residual mixture offers a marginal gain (+0.6%), showing that refinement itself is beneficial but insufficient. Introducing an adaptive policy, even without the stable residual structure (Direct Gating), yields a much larger improvement (+1.9% over the baseline), highlighting the critical role of our uncertainty-guided adaptation. Finally, combining both the **adaptive policy** and the **residual structure** in our full model unlocks the best performance (56.4%), outperforming the direct gating alternative by a significant 1.3 points.

This comprehensive analysis confirms that both core components of our DERM design—the principled, uncertainty-guided adaptive gate and the stable residual mixture formulation—are essential for achieving the superior performance and robustness of the GUIDE framework.

### B.13  SENSITIVITY TO TEMPERATURE SCALING

In Level ❷, the temperature $T$ scales the logits before the softmax operation, which influences the sharpness of expert distributions for uncertainty estimation. A higher temperature produces softer distributions, while a lower temperature leads to sharper ones. To validate the robustness of our framework to this hyperparameter, we conducted a sensitivity analysis on CIFAR-100-LT (IR=100)

by varying $T$ while keeping all other settings fixed. For each value of $T$, we report the mean and standard deviation of Top-1 accuracy over three independent runs with different random seeds.

As illustrated in Figure 8, the performance of GUIDE remains remarkably stable across a wide range of temperature values. The small error bars indicate low variance across independent runs, confirming the stability of our training procedure. While extremely high temperatures (e.g., $T > 4$) can cause a slight degradation by making expert distributions overly uniform and thus masking true uncertainty, the model achieves statistically indistinguishable, near-optimal performance for any $T$ within the standard range of [0.5, 2.0]. Our choice of $T = 1.0$ as a default is therefore empirically justified as a simple and robust setting that does not require dataset-specific tuning.

## C LIMITATIONS AND FUTURE WORK

While GUIDE demonstrates strong performance, we identify areas for future exploration. The two-step inference strategy, which relies on an initial prediction, poses a theoretical risk of mis-guidance. Our analysis in Appendix B.5 shows that GUIDE's learned policies are remarkably robust to this, often correcting initial errors. However, future work could explore more advanced inference-time strategies, such as self-consistency over top-k predictions, to further enhance robustness.

Furthermore, while our evaluation on skewed test distributions provides strong evidence of robustness to prior shifts, a more systematic validation on corruption-based benchmarks, such as CIFAR-100-C, could further delineate the generalization boundaries of our disentanglement approach. We hypothesize that the principled uncertainty handling in Level ❷ and the stable optimization in Level ❸ would confer significant advantages in such noisy, out-of-distribution scenarios. Verifying this remains a promising direction for future research.

