# OpenReview forum: "GUIDE: Gated Uncertainty-Informed Disentangled Experts for Long-tailed Recognition"
_ICLR.cc/2026/Conference — ICLR 2026 Poster_

### Official Review · Reviewer_LdLA · 2025-10-30

**Soundness:** 4
**Presentation:** 3
**Contribution:** 4
**Rating:** 8
**Confidence:** 3

**Summary:**

GUIDE aims to improve Long-Tailed Recognition problem by solving homogeneity collapse in multi-expert architectures. Three levels of disentanglement in the learning process is identified as the bottleneck and handled sequentially. **Representation-decision entanglement** is handled by two regularization losses which encourages diversity at feature level and predictive logits level separately. **Cause-symptom entanglement** is handled by decomposing epistemic and aleatoric uncertainty to guide the adaptation policy. **Learning-meta-learning entanglement** in meta learning is handled by using different scales of learning rate for two sets of parameters. Through experiments, GUIDE shows efficacy at medium and few shot classes. The ablations show the necessity and synergy between disentanglements.

**Strengths:**

1. The paper is well written with clarity and well-illustrated in general.
2. The insights that different levels of entanglements are novel. Especially the disentanglement for representation-decision establishes a new paradigm beyond traditional logit adjustment.
3. The necessity of most components in the contributions is examined in the ablation study. Hyperparameter sensitivity also looks stable.
4. Proof in the theorems are valid and clear.

**Weaknesses:**

1. Theorem 1 only suggests maximizing JSD, not mentioning minimizing the cosine similarity between the feature vectors. Even though the synergy between two competitive regularizers are shown in the ablation, the motivation of discouraging cosine similarity is heuristic.
2. The applied approach decomposition of AU and EU is classical and recently shown as entangled [1]. Using separate modules for AU and EU may even boost performance.
3. adaptive residual mixture is not ablated and there is no motivation
4. It is bit hard to understand the modle architecture from figure 2 as it contains both models, losses and optimization process.

[1] Mucsányi, B., Kirchhof, M., & Oh, S. J. (2024). Benchmarking uncertainty disentanglement: Specialized uncertainties for specialized tasks. Advances in neural information processing systems, 37, 50972-51038.

**Questions:**

1. What is motivation of designing the refinement strength a monotonically decreasing function with respect to its aleatoric uncertainty?
2. In sec 3.2, what are pathways? Why are x as the intput to the pathways?
3. How to get classwise EU and AU using class-level Exponential Moving Averages of the diagnosed uncertainties? as the entropy in (4) and (5) works on the whole predictive distribution.
4. In table 5, right, is the total uncertainty the entropy of average predictive distribution? What is the performance of using ambiguous signals like high training loss?

---

> ### Author Response · Authors · 2025-11-19
> **Response to Review LdLA**
>
> Dear Reviewer LdLA,
>
> We are immensely grateful for your exceptionally thorough and positive review. We are deeply encouraged by your recognition of our work's novelty, particularly the insights on hierarchical entanglement and the new paradigm our method establishes. Your expert feedback and astute questions have been instrumental in helping us refine and strengthen our paper. We have addressed each of your points below. **To facilitate your review, we have highlighted all significant revisions in the updated manuscript in blue.**
>
> > **1. On the Heuristic Motivation for Cosine Similarity. (For Weakness 1)**
>
> Your analysis is perfectly accurate. We agree that while Theorem 1 provides a direct motivation for maximizing JSD in the **prediction space**, the use of a cosine similarity loss in the **feature space** is a theoretically-inspired heuristic.
>
> Our motivation stems from our diagnosis of "representation-decision entanglement." We argue that only enforcing diversity at the decision level is insufficient, as powerful head-class gradients can still cause all experts' feature representations to collapse. To achieve a thorough disentanglement, we must apply diversity pressure at both levels simultaneously. The powerful synergy shown in **Table 5 (Left)** strongly validates this design; combining both losses yields a substantial **+4.6%** gain, far greater than the sum of their individual contributions, confirming this dual-pronged approach is critical.
>
> > **2. On the Entanglement of AU/EU Decomposition. (For Weakness 2)**
>
> Thank you for this insightful comment and for pointing us to this very relevant and recent work. We agree that this classic decomposition is not theoretically perfect and that these uncertainties can be entangled.
>
> Our argument is that, despite its theoretical limitations, this formulation provides a highly **effective diagnostic tool specifically for the long-tailed problem**, where distinguishing model ignorance (from data scarcity) and data ambiguity (from intra-class variance) is a key challenge. Our experiments provide strong evidence for its practical utility:
>
> - In **Table 5 (Right)**, our policy using this decomposition (56.4%) significantly outperforms one using the total, non-decomposed uncertainty (54.9%). This confirms that the decomposed signals, even if imperfect, contain crucial information that our framework successfully leverages.
>
> We will revise **Section 3.2** and **Appendix A.4** to cite this important paper and discuss this nuance, making our claims more precise.
>
> > **3. On Missing Ablation and Figure Clarity. (For Weakness 3 & 4)**
>
> We appreciate these constructive suggestions and have taken action to address them.
>
> * **New Ablation for Adaptive Residual Mixture**: This was an excellent point. We have conducted a new, comprehensive ablation study, now included in **Appendix B.12 (Table 16)**. We compare our full design against three alternatives: (1) no refinement, (2) a static residual mixture, and (3) an adaptive direct gating mechanism. The results confirm that both the **adaptive** nature of the gate and the **residual** structure are essential, with our full design outperforming the next best alternative by a significant 1.3 points.
>
> * **Figure 2 Clarity**: We agree that Figure 2 is dense. We will revise it for the final version to improve clarity by using a better layout and color-coding to distinguish the data flow, disentanglement levels, and optimization loops.

---

> ### Author Response · Authors · 2025-11-19
> **Response to Review LdLA**
>
> > **4: Motivation for refinement strength being a decreasing function of AU? (For Question 1)**
>
> Your intuition is correct. Our motivation is to prevent overfitting. High aleatoric uncertainty signifies irreducible data ambiguity or noise. Over-investing model capacity to refine features for such samples is not only ineffective but also risks fitting to noise. Therefore, a robust policy should reduce refinement for high-AU classes, a behavior theoretically guaranteed by our **Theorem 2**.
>
> > **5: What are "pathways" in Sec 3.2? (For Question 2)**
>
> Thank you for pointing out this ambiguity. "Pathways" refer to the network modules within the DERM: the **shared foundation pathway `F_found`** and the **expert-specific refinement pathways `F_refine,e`** (Eq. 6), which process the features extracted by the main backbone. We will clarify this in the revision.
>
> > **6: How to get class-wise EU and AU? (For Question 3)**
>
> This is a two-stage process detailed in **Algorithm 1 (lines 21-31)**.
> First, we compute sample-wise EU/AU on a validation set. Second, we aggregate these values per class and update a moving average (EMA) for each class's final EU and AU. This ensures the class-level estimates are stable and robust.
>
> > **7: Is total uncertainty in Table 5 the entropy of the average prediction? (For Question 4)**
>
> Yes, your understanding is exactly right. The "Agnostic (Total Uncert.)" policy uses `H(p̄_T(·|x))`, which is equivalent to `Epi_T(x) + Ale_T(x)`. Regarding using high training loss as a signal, that is indeed what many existing methods do.
>
> Our work argues this is a flawed "symptom-cause" signal, and the superior performance of our uncertainty-guided policy (56.4% vs. 54.9%) validates that our more principled diagnosis is more effective.
>
> ---
>
> We are truly grateful for your supportive and deeply insightful review, which has significantly improved the quality and completeness of our paper.
>
> Sincerely,
>
> The Authors of Submission 10916

---

> ### Author Response · Authors · 2025-11-26
> **Response to Review LdLA**
>
> Dear Reviewer LdLA,
>
> We hope this message finds you well.
>
> We have posted a detailed response to your review comments and updated the manuscript accordingly. We would greatly appreciate it if you could take a moment to check our rebuttal and let us know if our response has adequately addressed your concerns.
>
> We are happy to provide any further clarifications during the discussion period.
>
> Best regards,
>
> The Authors of Submission 10916

---

### Official Review · Reviewer_ARjA · 2025-11-01

**Soundness:** 2
**Presentation:** 3
**Contribution:** 3
**Rating:** 6
**Confidence:** 4

**Summary:**

This paper proposes GUIDE, a novel framework for solving the long-tail identification problem. The authors argue that existing multi-expert architectures suffer from three levels of entanglement: representation-decision entanglement, cause-symptom entanglement, and learning-meta-learning entanglement. GUIDE systematically addresses these problems through hierarchical entanglement resolution methods: (1) competitive specialization objectives achieve representation deentanglement, (2) adaptive policies based on uncertainty decomposition achieve policy deentanglement, and (3) dual-timescale optimization achieves optimization deentanglement. Experiments on five long-tail benchmark datasets demonstrate that GUIDE achieves a new SOTA.

**Strengths:**

1. The paper provides a novel and insightful diagnosis of long-tailed recognition challenges.
2. The module design is well-motivated. The uncertainty decomposition, though lacking rigorous justification, provides a practical heuristic for adaptive refinement that works in practice.
3. The experimental evaluation is thorough and convincing.

**Weaknesses:**

### 1. **Overstated Connection Between JSD Maximization and Performance Improvement**

The paper's Theorem 1(b) claims that "maximizing JSD serves to tighten this performance bound," but this relationship is **not rigorously proven**. The mathematical derivation shows:

-  **Problematic**: The gap in Jensen's bound ≠ JSD
-  **Unproven**: Maximizing JSD → Tightening the bound

The authors acknowledge in the proof that *"the gap in the Jensen bound is not strictly equal to JSD"* and that maximizing JSD *"empirically encourages"* diversity. This is an **empirical heuristic**, not a theoretical guarantee. The authors can provide rigorous proof, or explicitly redefine it as an empirical observation rather than a theoretical contribution.

### 2. **Lack of Theoretical Foundation for Uncertainty Decomposition**

The epistemic/aleatoric uncertainty decomposition (Equations 4-5) is presented as a principled approach but lacks theoretical justification:
```
Aleatoric: AleT(x) = (1/E)ΣH(pe,T(·|x))
Epistemic: EpiT(x) = H(p̄T(·|x)) - AleT(x)
```

**Issues**:
- No proof that this decomposition correctly identifies "model ignorance" vs "data ambiguity"
- No theoretical guarantee that high epistemic uncertainty → need more refinement
- No validation that high aleatoric uncertainty → avoid over-refinement

While the appendix mentions this is a *"standard and widely-accepted definition in the Bayesian deep learning community,"* this is **community convention**, not mathematical fact. The entire Level ❷ policy is built on this unproven assumption. The authors need to clarify that this is an empirical design choice and provide empirical validation (e.g., showing the correlation between these quantities and actual model/data properties), or provide theoretical justification for why this decomposition is suitable for this task.

### 3. **Lack of related work**

MDCS: More diverse experts with consistency self-distillation for long-tailed recognition (ICCV2023) is also a method that uses diversity experts and ensemble learning in long-tail recognition, and it needs to be compared and discussed.

**Questions:**

How do the inference efficiency and training memory consumption of this method compare to other methods?

---

> ### Author Response · Authors · 2025-11-19
> **Response to Review ARjA**
>
> Dear Reviewer ARjA,
>
> We are sincerely grateful for your rigorous and insightful review. We are encouraged that you recognized our "novel and insightful diagnosis" and "thorough and convincing" experimental evaluation. Your expert feedback on our theoretical framing has been invaluable, and we have carefully revised our manuscript to enhance its precision and rigor. We address your points below. **To facilitate your review, we have highlighted all significant revisions in the updated manuscript in blue.**
>
> > **1. On the Connection Between JSD and Performance. (For Weakness 1)**
>
> Thank you for your sharp observation. We agree that our original claim of a direct theoretical guarantee is inaccurate. Per your suggestion, we have revised our manuscript to more accurately frame this relationship.
>
> * **Clarification in Manuscript**: We have revised **Section 3.1** to clarify that while the performance bound is not strictly equal to JSD, maximizing JSD serves as a **theoretically-motivated objective**. It acts as an effective proxy for encouraging expert diversity, a well-established principle in ensemble learning for improving performance. We have repositioned this from a "theoretical proof" to a "theoretically-inspired and empirically successful design choice."
>
> * **Empirical Evidence**: The effectiveness of this objective is validated in our ablation study (**Table 5, Left**), where introducing JSD maximization (`+ L_div only`) alone brings a significant performance gain of +1.7% over the baseline.
>
> > **2. On the Theoretical Foundation of Uncertainty Decomposition. (For Weakness 2)**
>
> This is a crucial point, and we thank you for pushing us to provide stronger justification. We acknowledge that this decomposition is an empirically-grounded formulation from the Bayesian deep learning community, not a first-principles theorem. To address your request for validation, we now explicitly provide a multi-faceted empirical proof within our paper to demonstrate its effectiveness for our task.
>
> * **Quantitative Validation**: Our ablation study on gating policies (**Table 5, Right**) provides direct quantitative evidence. The `GUIDE Policy` (56.4%), which uses the decomposed uncertainty, significantly outperforms the `Agnostic (Total Uncert.)` policy (54.9%) that does not. This +1.5% gain confirms that **decomposing uncertainty provides a meaningful and effective signal for adaptation**.
>
> * **Qualitative & Causal Validation**: Our visualizations and case studies provide strong corroborating evidence that the decomposition correctly identifies the underlying causes of difficulty:
>     * **Figure 5 in Appendix B.7** clearly shows a strong correlation where data-scarce tail classes (high model ignorance) exhibit high epistemic uncertainty, while data-rich but ambiguous head classes exhibit high aleatoric uncertainty.
>     * **Table 13 in Appendix B.8** provides a concrete case study showing that a rare class receives a high refinement gate value (0.92), while an ambiguous class receives a low one (0.31).
>
> We have revised **Section 3.2** to state the empirical nature of this definition and added a new paragraph to **Section 4.3** to explicitly highlight these empirical validations, directly linking our results to your query.
>
> > **3. On Missing Related Work. (For Weakness 3)**
>
> Thank you for pointing out the omission of MDCS (ICCV 2023). This was an oversight on ourpart. We have taken the following actions:
>
> * **Discussion**: We have cited MDCS and included a discussion of MDCS in **Section 2 (Related Work)**. We clarify that while MDCS also promotes diversity implicitly, our hierarchical framework provides a more comprehensive solution by explicitly addressing a broader cascade of entanglements, from representation to optimization.
>
> * **Performance Comparison**: We have added the results of MDCS to our main comparison in **Table 1** to ensure a comprehensive evaluation against this strong baseline.
>
> > **4. On Computational Cost. (For Question 1)**
>
> We agree this is an important practical consideration. In response to this and a similar query from Reviewer 8uEt, we have added a new **Appendix B.9** with a detailed cost-benefit analysis.
>
> * **New Table 14** compares GUIDE's computational costs (FLOPs, memory, time) against strong multi-expert baselines. The analysis shows that GUIDE's moderate additional cost yields a transformative **+8.0 absolute point gain in Few-shot accuracy**, validating it as a highly efficient investment for tackling the core LTR challenge.
>
> ---
>
> We are confident that these revisions have fully addressed your concerns and have made our paper stronger and more rigorous. Thank you once again for your invaluable feedback.
>
> Sincerely,
>
> The Authors of Submission 10916

---

> ### Author Response · Authors · 2025-11-26
> **Response to Review ARjA**
>
> Dear Reviewer ARjA,
>
> We hope this message finds you well.
>
> We have posted a detailed response to your review comments and updated the manuscript accordingly. We would greatly appreciate it if you could take a moment to check our rebuttal and let us know if our response has adequately addressed your concerns.
>
> We are happy to provide any further clarifications during the discussion period.
>
> Best regards,
>
> The Authors of Submission 10916

---

### Official Review · Reviewer_Vmze · 2025-11-02

**Soundness:** 3
**Presentation:** 3
**Contribution:** 2
**Rating:** 6
**Confidence:** 4

**Summary:**

This paper introduces GUIDE, a novel framework conceived from the philosophy of Hierarchical Disentanglement.

**Strengths:**

A novel framework conceived from the philosophy of Hierarchical Disentanglement is proposed. In addition, this paper systematically addresses these issues at distinct levels.

**Weaknesses:**

The authors only verify the method on computer vision datasets. If possible, the proposed method is suggested on more long-tailed datasets (e.g., MEET) from more domains.

**Questions:**

Please consider to improve the experimental analysis on more datasets.

---

> ### Author Response · Authors · 2025-11-19
> **Response to Review Vmze**
>
> # Response to Review Vmze
>
> Dear Reviewer Vmze,
>
> We thank you for your thorough review and for recognizing the novelty of our Hierarchical Disentanglement framework.
> We fully agree with your excellent suggestion that demonstrating GUIDE's efficacy beyond standard CV datasets is crucial for validating its generalizability. **To facilitate your review, we have highlighted all significant revisions in the updated manuscript in blue.**
>
> Inspired by your feedback, we have not only extended our evaluation to new domains but have done so in a manner that directly stress-tests the core long-tailed learning capabilities of our framework. These new, principled experiments are now detailed in the revised **Appendix B.4, "Extended Evaluation on Diverse and Pathological Long-Tailed Scenarios."**
>
> > **1. A New Benchmark: Long-Tailed Fine-Grained Geospatial Recognition (MEET-LT).**
>
> Inspired by your suggestion to use the MEET dataset, we went a step further and propose a new challenging long-tailed benchmark, **MEET-LT**. We constructed a long-tailed version of MEET with a severe **imbalance ratio of IR=100** using the standard exponential decay protocol, creating a scenario that is both domain-novel and directly relevant to our paper's core problem.
>
> * As detailed in the new **Table 9 (Appendix B.4)**, we present a comprehensive performance evaluation against a suite of strong baselines. GUIDE achieves an overall accuracy of **75.5%**.
>
> * Crucially, the source of this significant **+4.0 point gain** over the strong BCL baseline (71.5%) lies in its transformative impact on the most challenging classes. GUIDE boosts the **Few-shot accuracy to 57.5%**, a remarkable **+7.5 point absolute improvement**. This directly validates that GUIDE's core mechanisms are exceptionally effective at learning from scarce data in new, specialized domains.
>
> > **2. Performance on Pathological Scientific Graph Data (ZincFluor).**
>
> To further demonstrate GUIDE's versatility on a non-visual, non-Euclidean data modality, we evaluated it on the ZincFluor dataset for molecular property prediction, which features a pathological long-tail ($T=137.54$).
>
> * The results, presented in the new **Table 10**, show that GUIDE achieves state-of-the-art performance. On the critical **Tail Top-2 accuracy** metric, our method achieves **67.7%**, substantially outperforming all other established LTR methods, including the strong contrastive learning method BCL (59.6%).
>
> * This result confirms GUIDE’s robustness and capability in high-stakes scientific discovery scenarios where data is non-Euclidean and performance on the extreme tail is paramount.
>
> In summary, our new extensive evaluations on **long-tailed remote sensing (MEET-LT)** and **pathological molecular graph analysis (ZincFluor)** provide direct and powerful evidence for the broad applicability of our GUIDE framework. We have shown that our hierarchical disentanglement principle is a potent, general-purpose solution for a wide range of challenging long-tailed recognition problems.
>
> ---
>
> We are very grateful for your feedback, which inspired us to significantly strengthen the experimental validation in our paper. We hope you will find these new, targeted results compelling in your final assessment.
>
> Sincerely,
>
> The Authors of Submission 10916

---

> ### Author Response · Authors · 2025-11-26
> **Response to Review Vmze**
>
> Dear Reviewer Vmze,
>
> We hope this message finds you well.
>
> We have posted a detailed response to your review comments and updated the manuscript accordingly. We would greatly appreciate it if you could take a moment to check our rebuttal and let us know if our response has adequately addressed your concerns.
>
> We are happy to provide any further clarifications during the discussion period.
>
> Best regards,
>
> The Authors of Submission 10916

---

### Official Review · Reviewer_8uEt · 2025-11-03

**Soundness:** 3
**Presentation:** 3
**Contribution:** 3
**Rating:** 6
**Confidence:** 4

**Summary:**

This paper addresses the problem of long tailed recognition (LTR) in multi-expert architectures via a so-called hierarchical disentanglement (GUIDE). This is done by introducing specific mechanisms that address problems such as entaglement causing homogeneity collapse, entanglement of policy-making from ambiguous signals by decomposing epistemic/aleatoric uncertainty with a dynamic refinement module, and introduces an approach for stabilizing convergence via different timescale updates. Experiments on several dataset demonstrate improvements. Please see below for discussion.

**Strengths:**

Below I list some of the main strengths of the paper:
- the three leveled framework is theoretically motivated
- results appear to improve over state of the art
- particularly strong improvement on few-shot classes
- comprehensive evaluation (e.g. distribution shift)
- ablation studies validate the contribution of each component separately, and the impact of combining two components leading to further improvements.

**Weaknesses:**

- Not sufficient discussion on complexity and convergence rates
- Discussion on how hyperparameters (inc learning rates and various λ) affect guarantees
- hierarchical disentanglement can mostly be considered as a smart combination of existing approaches
- would be interesting to include more analysis of failure cases
- improve for clarity - be clear on problems that are existing challenges and not new, and on what is proven theoretically vs empirically validated

**Questions:**

- some more ellaboration on design choices; e.g. why in fig 3B only marginal gains are achieved with E=4? Why residual gating? (eq. 6)
- B.4 - robustness to mis-guided refinement: 21.7% of samples were correctly classified after the refinement step, even when guided by incorrect proxy label. What about the rest? what is the net effect?
- analysis on computational cost, memory footprint
- please see also above

---

> ### Author Response · Authors · 2025-11-19
> **Response to Review 8uEt**
>
> Dear Reviewer 8uEt,
>
> We sincerely thank you for your detailed and insightful review. We are greatly encouraged that you recognized our work's theoretical motivation, strong empirical results, and comprehensive evaluations. Your feedback has been invaluable in helping us improve the manuscript. We have carefully addressed each of your points below, with corresponding additions and clarifications in the revised appendix. **To facilitate your review, we have highlighted all significant revisions in the updated manuscript in blue.**
>
>
> > **1. On Computational Complexity, Convergence, and Cost. (For Weakness 1 & Q3)**
>
> We agree that a discussion on computational overhead is crucial. To address this, we have added a comprehensive analysis in a new section, **Appendix B.9**.
>
> * **Cost-Benefit Analysis:** The new **Table 14 (in Appendix B.9)** provides a detailed cost-benefit analysis against several strong multi-expert baselines. It shows that while GUIDE's training costs are moderately higher (e.g., +12 min train time vs. BalPoE), the real-world inference latency increases by only ~2ms. This minimal extra cost is a highly effective investment, delivering a transformative **+8.0 absolute point gain** in few-shot accuracy over the next best baseline on this challenging task.
> * **Convergence:** The new **Figure 6 (in Appendix B.9)** illustrates GUIDE's stable and efficient convergence. This validates our two-timescale optimization and shows that our superior performance is achieved without compromising training stability.
>
>
> > **2. On Hyperparameters and Design Choices. (For Weakness 2 & Q1)**
>
> Thank you for these important questions. We have now provided detailed sensitivity analyses and clarifications.
>
> * **Hyperparameter Sensitivity:**
>     * For the diversity weights ($\lambda$), as shown in **Figure 3(a)** in the main paper, our model is highly robust.
>     * For the meta-policy learning rate ($\eta_\phi$), we have added a new ablation study in **Appendix B.11** (Table 15). The results, consistent with TTSA theory, demonstrate that performance remains near-optimal and stable across a range of small values. Our default choice of 1e-4 is situated within this robust range.
>
> * **Design Choices:**
>
>     * **Number of Experts (E=4):** Your observation is astute. As shown in **Figure 3(b)**, increasing the number of experts from 1 to 3 yields dramatic performance gains. The marginal returns at E=4 indicate that E=3 already provides sufficient diversity. Therefore, we chose E=3 as it represents an optimal trade-off between performance and computational efficiency.
>
>     * **Residual Gating (Eq. 6):** Its key advantage, detailed in **Appendix A.4**, is enabling a smooth interpolation between the base representation (when gate value $g \approx 0$) and the fully refined one ($g \approx 1$). This mechanism is significantly more stable and flexible than a direct feature switch.

---

> ### Author Response · Authors · 2025-11-19
> **Response to Review 8uEt**
>
> > **3. On Novelty and Clarity of Contribution. (For Weakness 3 & 5)**
>
> We agree with your insightful characterization and have revised our manuscript for clarity. We acknowledge that some components are based on established concepts. Our core novelty lies not in inventing these components, but in **(1) our novel diagnosis of the 'dependency chain of entanglement'** and **(2) the proposal of a principled, hierarchical framework** to resolve it. The synergy demonstrated in our ablations (**Table 4**) validates that our framework's contribution is far greater than the sum of its parts. Per your suggestion, we have revised the text to more clearly distinguish our novel diagnoses from existing challenges, and theoretical proofs from empirical validations.
>
>
>
> > **4. On Deeper Analysis of Failure Cases and Mis-guidance. (For Weakness 4 & Q2)**
>
> These are excellent suggestions for deeper analysis, which we have now incorporated.
>
> * **Challenging Cases Analysis:** We have added a new **Appendix B.10**, "Qualitative Analysis of Challenging Fine-Grained Examples." It first presents a success case where GUIDE correctly distinguishes between species that are confusing to humans. It then analyzes failure modes, which are typically confined to scenarios of extreme fine-grained similarity that are also difficult for human experts (see new **Figure 7**).
>
> * **"Mis-guided Refinement":** We apologize for the lack of clarity and thank you for the opportunity to elaborate. The 21.7% figure is a **strong positive result** demonstrating the robustness of our learned policies. This analysis focused on the subset of samples that were initially misclassified (i.e., with an initial accuracy of 0%). Our refinement step successfully **corrected 21.7% of these errors**, even when guided by the initial *incorrect* proxy label. This proves that guidance from a semantically related (though wrong) class is more beneficial than no adaptation at all. The net effect is the significant +3.3% overall accuracy improvement shown in Table 11. We have revised **Appendix B.5** to make this explicit.
>
> ---
>
> We believe these revisions and extensive new analyses in the appendix have thoroughly addressed your concerns and have significantly strengthened our paper. Thank you once again for your meticulous and valuable guidance.
>
> Sincerely,
>
> The Authors of Submission 10916

---

> ### Author Response · Authors · 2025-11-26
> **Response to Review 8uEt**
>
> Dear Reviewer 8uEt,
>
> We hope this message finds you well.
>
> We have posted a detailed response to your review comments and updated the manuscript accordingly. We would greatly appreciate it if you could take a moment to check our rebuttal and let us know if our response has adequately addressed your concerns.
>
> We are happy to provide any further clarifications during the discussion period.
>
> Best regards,
>
> The Authors of Submission 10916

---

> > ### Comment · Reviewer_8uEt · 2025-11-27
> >
> > I would like to thank the reviewers for providing detailed responses to my comments, clarifying several points and diligently adding new material to the paper. My review remains positive for the paper.

---

### Author Response · Authors · 2025-12-01
**Summary of Revisions and Reviewer Status**

Dear Area Chair,

First and foremost, we would like to express our sincere gratitude for your time and effort in handling our submission. We understand that taking over the assignment at this stage requires significant energy, and we deeply appreciate your dedication to the review process.
To facilitate your efficient evaluation of our work, we provide a summary of the current reviewer status and the major improvements made during the rebuttal.

### Current Reviewer Status
We have actively engaged with all reviewers. The current status is positive:
*   **Reviewer LdLA (Score: 8):** Gave a strong acceptance rating, praising the "excellent contribution" and "novel insights."
*   **Reviewer 8uEt (Score: 6):** Following our response (cost-benefit analysis and convergence plots), this reviewer has replied stating that their **review remains positive**.
*   **Reviewers Vmze & ARjA (Score: 6):** We have provided comprehensive new experiments (new benchmarks) and theoretical clarifications to address their constructive comments. We believe our revisions have effectively resolved their concerns.

### Key Revisions and Technical Highlights
We have significantly strengthened the paper in three key areas, incorporating extensive new experiments in the Appendix:

**1. Proven Generalizability on Diverse Domains (Addressing Vmze)**
*   **Critique:** Requested verification beyond standard CV datasets.
*   **Resolution:** We conducted principled experiments on two new challenging benchmarks in **Appendix B.4**:
    *   **MEET-LT (Remote Sensing):** A fine-grained geospatial dataset (IR=100).
    *   **ZincFluor (Molecular Graph):** A pathological long-tailed scientific discovery dataset ($T=137.54$).
*   **Outcome:** GUIDE achieves **75.5%** accuracy on MEET-LT (+4.0% over BCL) and **67.7%** Tail Top-2 accuracy on ZincFluor. This empirically proves GUIDE's efficacy across diverse modalities.

**2. Cost-Benefit & Convergence Analysis (Addressing 8uEt, ARjA)**
*   **Critique:** Requested analysis on computational complexity and training stability.
*   **Resolution:** We added a detailed analysis in **Appendix B.9**.
*   **Outcome:** We demonstrated that while training cost is moderately higher, inference latency increases by only ~2ms. This investment yields a transformative **+8.0% absolute gain** in Few-shot accuracy on CIFAR-100-LT. New plots also confirm stable convergence via our two-timescale optimization.

**3. Theoretical Rigor & Ablations (Addressing ARjA, LdLA)**
*   **Critique:** Requested clarification on JSD theory and motivation for specific module designs.
*   **Resolution:**
    *   Revised Section 3.1 to accurately frame JSD maximization.
    *   Validated the empirical efficacy of uncertainty decomposition (Epistemic vs. Aleatoric) in **Table 5**.
    *   Added comprehensive ablations for the *Adaptive Residual Mixture* in **Appendix B.12**.
*   **Outcome:** Results confirm that the adaptive residual design outperforms direct gating by 1.3 points, and the uncertainty-guided policy significantly outperforms agnostic baselines.

### Conclusion
We have successfully addressed the specific concerns of the reviewers with solid empirical evidence and rigorous analysis. With strong support from Reviewer LdLA and confirmed positivity from Reviewer 8uEt, we are confident that GUIDE represents a robust and novel contribution to the ICLR community.

Sincerely,

The Authors of Submission 10916

---

### Meta-Review · Area_Chair_3t8f · 2026-01-10

**Summary:**

This paper proposes Gated Uncertainty-Informed Disentangled Experts(GUIDE) for long-tailed recognition. The core diagnosis is that multi-expert systems fail not for one reason but because of a cascade of entanglements—in representation, in decision policy, and in optimization. GUIDE addresses these with 1) competitive specialization to force expert diversity, 2) uncertainty-aware dynamic refinement to separate model ignorance from data ambiguity, and 3) two-timescale optimization to stabilize meta-learning. Across five benchmarks (ImageNet-LT, iNaturalist, CIFAR-10/100-LT, Places-LT) and new domains added in rebuttal, GUIDE establishes a new state of the art, especially on few-shot classes

Here are the main concerns and how the rebuttal resolved them.

1) Theory overstated vs. empirical design.
Reviewers (ARjA, 8uEt) objected that the original text claimed theoretical guarantees (e.g., via JSD) that were not strictly proven. The rebuttal explicitly corrected the framing, repositioning these components as theoretically-motivated but empirically validated. New ablations showed that the diversity objective and uncertainty-guided policy each yield measurable gains on their own, and together deliver much larger improvements.

2) Validity of uncertainty-guided refinement.
Skepticism remained about whether separating epistemic vs. aleatoric uncertainty was principled or useful. The authors added quantitative policy ablations showing that the decomposed-uncertainty policy outperforms an agnostic “total-uncertainty” gate, and qualitative and class-level analyses showing that rare classes receive strong refinement while ambiguous head classes are appropriately suppressed.

3) Missing ablations and design justification.
Reviewers (LdLA, 8uEt) asked for ablations of residual gating, number of experts, and refinement strength. The rebuttal added a full adaptive-mixture ablation (static vs. adaptive vs. no refinement), expert-count sweeps showing saturation at E≈3, and a stability-motivated explanation for reducing refinement under high aleatoric uncertainty.

4) Generalization beyond standard CV benchmarks.
Vmze requested evidence outside ImageNet-like settings. The authors added two new domains: a long-tailed remote-sensing benchmark (MEET-LT) and a molecular graph dataset (ZincFluor). GUIDE achieved large tail-class gains on both, demonstrating modality-agnostic effectiveness.

5) Efficiency and convergence.
8uEt and ARjA asked about cost and stability. A new cost–benefit and convergence analysis showed only ~2ms inference overhead and stable two-timescale training, while delivering large few-shot accuracy gains.

Overall, the rebuttal corrected theoretical overclaims, added missing ablations, validated the uncertainty policy, expanded domains, and quantified cost and stability. One reviewer already raised their score to a strong accept, and others confirmed their positivity after seeing the revisions.

Recommendation: Accept.

**Reviewer Concerns:**

Here are the main concerns and how the rebuttal resolved them.

1) Theory overstated vs. empirical design.
Reviewers (ARjA, 8uEt) objected that the original text claimed theoretical guarantees (e.g., via JSD) that were not strictly proven. The rebuttal explicitly corrected the framing, repositioning these components as theoretically-motivated but empirically validated. New ablations showed that the diversity objective and uncertainty-guided policy each yield measurable gains on their own, and together deliver much larger improvements.

2) Validity of uncertainty-guided refinement.
Skepticism remained about whether separating epistemic vs. aleatoric uncertainty was principled or useful. The authors added quantitative policy ablations showing that the decomposed-uncertainty policy outperforms an agnostic “total-uncertainty” gate, and qualitative and class-level analyses showing that rare classes receive strong refinement while ambiguous head classes are appropriately suppressed.

3) Missing ablations and design justification.
Reviewers (LdLA, 8uEt) asked for ablations of residual gating, number of experts, and refinement strength. The rebuttal added a full adaptive-mixture ablation (static vs. adaptive vs. no refinement), expert-count sweeps showing saturation at E≈3, and a stability-motivated explanation for reducing refinement under high aleatoric uncertainty.

4) Generalization beyond standard CV benchmarks.
Vmze requested evidence outside ImageNet-like settings. The authors added two new domains: a long-tailed remote-sensing benchmark (MEET-LT) and a molecular graph dataset (ZincFluor). GUIDE achieved large tail-class gains on both, demonstrating modality-agnostic effectiveness.

5) Efficiency and convergence.
8uEt and ARjA asked about cost and stability. A new cost–benefit and convergence analysis showed only ~2ms inference overhead and stable two-timescale training, while delivering large few-shot accuracy gains.

**Reviewer Scores:**

LdLA: already 8 (strong accept) — remains strongly positive after new ablations and clarifications.

8uEt: 6 → likely 7 — confirmed post-rebuttal that their review “remains positive” after cost, convergence, and design-choice analyses.

Vmze: 6 → likely 7 — domain-generalization concern resolved by MEET-LT and ZincFluor experiments.

ARjA: 6 → likely 7 — theoretical overclaims were corrected, MDCS added, uncertainty policy validated, and cost analyzed.

---

### Decision · Program_Chairs · 2026-01-26

Accept (Poster)